# A homozygous missense variant in *CACNB4* encoding the auxiliary calcium channel beta4 subunit causes a severe neurodevelopmental disorder and impairs channel and non-channel functions

Pierre Coste de Bagneaux[1◉], Leonie von Elsner[2◉], Tatjana Bierhals[2◉], Marta Campiglio[1], Jessika Johannsen[3], Gerald J. Obermair[1,4], Maja Hempel[2], Bernhard E. Flucher[1‡]*, Kerstin Kutsche[2‡]*

1 Department of Physiology and Medical Physics, Medical University Innsbruck, Innsbruck, Austria,
2 Institute of Human Genetics, University Medical Center Hamburg-Eppendorf, Hamburg, Germany,
3 Childrens Hospital, University Medical Center Hamburg-Eppendorf, Hamburg, Germany, 4 Division Physiology, Karl Landsteiner University of Health Sciences, Krems, Austria

◉ These authors contributed equally to this work.
‡ These authors also contributed equally to this work.
* bernhard.e.flucher@i-med.ac.at (BEF); kkutsche@uke.de (KK)

## Abstract

P/Q-type channels are the principal presynaptic calcium channels in brain functioning in neurotransmitter release. They are composed of the pore-forming $Ca_V2.1$ $\alpha_1$ subunit and the auxiliary α2δ-2 and $\beta_4$ subunits. $\beta_4$ is encoded by *CACNB4*, and its multiple splice variants serve isoform-specific functions as channel subunits and transcriptional regulators in the nucleus. In two siblings with intellectual disability, psychomotor retardation, blindness, epilepsy, movement disorder and cerebellar atrophy we identified rare homozygous variants in the genes *LTBP1*, *EMILIN1*, *CACNB4*, *MINAR1*, *DHX38* and *MYO15* by whole-exome sequencing. *In silico* tools, animal model, clinical, and genetic data suggest the p.(Leu126-Pro) *CACNB4* variant to be likely pathogenic. To investigate the functional consequences of the *CACNB4* variant, we introduced the corresponding mutation L125P into rat $\beta_{4b}$ cDNA. Heterologously expressed wild-type $\beta_{4b}$ associated with GFP-$Ca_V1.2$ and accumulated in presynaptic boutons of cultured hippocampal neurons. In contrast, the $\beta_{4b}$-L125P mutant failed to incorporate into calcium channel complexes and to cluster presynaptically. When co-expressed with $Ca_V2.1$ in tsA201 cells, $\beta_{4b}$ and $\beta_{4b}$-L125P augmented the calcium current amplitudes, however, $\beta_{4b}$-L125P failed to stably complex with $\alpha_1$ subunits. These results indicate that p.Leu125Pro disrupts the stable association of $\beta_{4b}$ with native calcium channel complexes, whereas membrane incorporation, modulation of current density and activation properties of heterologously expressed channels remained intact. Wildtype $\beta_{4b}$ was specifically targeted to the nuclei of quiescent excitatory cells. Importantly, the p.Leu125Pro mutation abolished nuclear targeting of $\beta_{4b}$ in cultured myotubes and hippocampal neurons. While binding of $\beta_{4b}$ to the known interaction partner PPP2R5D (B56δ) was not affected by the mutation, complex formation between $\beta_{4b}$-L125P and the neuronal TRAF2 and NCK

**Data Availability Statement:** All relevant data are within the manuscript and its Supporting Information files.

**Funding:** This study was supported by a grant from the Deutsche Forschungsgemeinschaft (KU 1240/6-2) to KK, and grants from the Austrian Science Fund (FWF) T855 to MC, F4415 to GJO, and P30402 and W1101 to BEF. The funders had no role in study design, data collection and analysis, decision to publish, or preparation of the manuscript.

**Competing interests:** The authors have declared that no competing interests exist.

interacting kinase (TNIK) seemed to be disturbed. In summary, our data suggest that the homozygous *CACNB4* p.(Leu126Pro) variant underlies the severe neurological phenotype in the two siblings, most likely by impairing both channel and non-channel functions of $\beta_{4b}$.

## Author summary

Neurodevelopmental disorders encompass a broad spectrum of neurological and psychiatric conditions and are caused by mutations in many different genes. For example, mutations in genes encoding voltage-gated calcium channels have been linked to various diseases of the nervous system in humans and mice. Voltage-gated calcium channels are critical regulators of the synaptic communication between neurons, of processes involved in learning and memory, and of activity-dependent gene expression. Here we report a disease-associated mutation on both copies of the *CACNB4* gene encoding an auxiliary $\beta_4$ subunit of the chief presynaptic calcium channel in the brain. Two siblings with a severe neurodevelopmental disorder carry the homozygous *CACNB4* mutation causing an amino acid substitution known to disrupt the folding of the calcium channel $\beta_4$ subunit. We demonstrate that this amino acid change abolished the incorporation of the $\beta_4$ subunit into channel complexes in the synapse, as well as $\beta_4$'s ability to translocate into the cell nucleus, and to complex with $\alpha_1$ channel subunits and a neuronal scaffolding protein. The combined evidence from our genetic and functional analysis suggests that dysfunction of both $\beta_4$ subunit channel and non-channel functions underlies the severe neurological phenotype in the two siblings. We therefore identified *CACNB4* as a neurodevelopmental disease gene.

## Introduction

P/Q-type channels are the principal presynaptic calcium channels functioning in rapid neurotransmitter release [1–3]. In brain, P/Q-type calcium channels are largely composed of the pore-forming $Ca_V2.1$ $\alpha_1$ and the auxiliary $\alpha2\delta$-2 and $\beta_4$ subunits [4–6]. The auxiliary $\alpha2\delta$ and $\beta$ subunits regulate the amplitude, kinetics, and voltage-dependence of calcium currents by enhancing functional membrane expression and modulating gating properties of high-voltage-gated calcium channels [7]. Certain auxiliary $\alpha2\delta$ and $\beta$ isoforms have channel-independent functions in synapse formation and activity-dependent transcriptional regulation, respectively [8–11]. Mutations in genes encoding each of these subunits have been associated with neurological disease [10, 12–15]. Similarly, loss-of-function mutations and knockout of $Ca_V2.1$, $\alpha2\delta$-2, or $\beta_4$ in mice cause severe neurological phenotypes including migraine, epilepsy, and ataxia [16–22]. The gene encoding $\beta_4$, *CACNB4*, is primarily expressed in brain, and its expression levels increase during development [5, 23–25]. *CACNB4* undergoes extensive alternative splicing, and the resulting variants ($\beta_{4a}$, $\beta_{4b}$, $\beta_{4c}$, $\beta_{4e}$) partly display distinct subcellular localizations and functions [9, 26, 27]. In neurons, the $\beta_{4e}$ isoform is primarily presynaptic, whereas $\beta_{4a}$ and $\beta_{4b}$ play dual roles in channel modulation and gene regulation. In electrically inactive neurons $\beta_{4b}$ and, to a lesser extent $\beta_{4a}$, are targeted to nuclei where they are involved in the regulation of gene transcription [26, 28]. The $\beta_{4b}$ variant has been reported to interact with the regulatory subunit of phosphatase 2A, Ppp2r5d (alternative name B56$\delta$), and with the transcription factor thyroid receptor $\alpha$ [29], and heterologous overexpression of $\beta_{4b}$ caused differential expression of genes involved in cell proliferation [30]. Calcium channel $\beta$ subunits

contain a tandem src homology 3 (SH3) and guanylate kinase (GK) module [9, 31]. Amino acid changes disturbing the intramolecular SH3-GK interaction affect β's roles in channel modulation, nuclear targeting, and its association with transcriptional regulators [29, 31, 32].

Consistent with the high expression of $\beta_4$ in cerebellar Purkinje and granule cells, $\beta_4$ null mutant mice (*lethargic*) display an autosomal recessive neurological disease [21] with ataxia, paroxysmal dyskinesia, and absence seizures [19, 33]. Neurons of *lethargic* mice show decreased P/Q-type calcium currents and excitatory synaptic transmission. On a network level, some of the defects in *lethargic* brains resembled those in $Ca_V2.1$-null mice, whereas others were specific to $\beta_4$ null mice, suggesting that mechanisms other than deficient P/Q-type currents cause the severe motor deficits [34]. Thus, it has been suggested that the lack of specific channel-independent functions of $\beta_4$ in activity-dependent gene regulation may be causal for the *lethargic* phenotype [26, 29, 34].

In humans, heterozygous variants in *CACNB4* (MIM: 601949) have been associated with different neurological phenotypes: a female patient with juvenile myoclonic epilepsy (JME) had the nonsense variant c.1444C>T/p.(Arg482*) (MIM: 607682), two members of a family displaying idiopathic generalized epilepsy with rare generalized tonic-clonic seizures carried the non-synonymous *CACNB4* variant c.311G>T/p.(Cys104Phe), and in another family five individuals affected by episodic ataxia (MIM: 613855) as well as two healthy family members showed the p.(Cys104Phe) variant [15]. The heterozygous *CACNB4* variant c.1403G>A/p.(Arg468Gln) has been suggested to worsen the neurological disorder in a patient with a pathogenic *SCN1A* mutation by increasing calcium channel current densities [35].

Here we report two patients, a 15-year-old boy and his 22-year-old sister, with severe intellectual disability, seizures, visual impairment, and dystonic and athetoid movements, carrying the homozygous *CACNB4* missense variant c.377T>C/p.(Leu126Pro). Functional analysis of mutant $\beta_4$ employing heterologous and homologous expression systems revealed striking effects of the amino acid substitution on calcium channel complex formation and $\beta_4$'s nuclear functions. Both of which, separately or together, could explain the severe neurological disease in brother and sister.

## Results

### Identification of the homozygous *CACNB4* missense mutation c.377C>T/p.(Leu126Pro) in two siblings with a severe neurodevelopmental disorder

We performed trio or duo whole-exome sequencing (WES) in a total of 390 pediatric subjects with a neurodevelopmental disorder as described previously [36, 37]. Analysis of WES data was performed according to an X-linked, autosomal recessive and autosomal dominant inheritance model, the latter with a *de novo* mutation in the affected child. WES in a male patient (patient 1) and his first-degree consanguineous healthy parents identified a total of 12 rare homozygous variants [with a minor allele frequency (MAF) <0.1% in the population databases dbSNP138, 1000 Genomes Project, Exome Variant Server, ExAC and gnomAD browsers and no homozygotes in ExAC and gnomAD browsers]. WES did not detect a *de novo* variant in patient 1. Trio-WES data were not filtered for X-linked variants as patient 1 had a sister (patient 2) who was similarly affected (S1 Text and Table 1). The 15-year-old patient 1 and the 22-year-old patient 2 were affected by severe intellectual disability, seizures, visual impairment, dystonic and athetoid movements (S1 Text and Table 1). Brain imaging revealed atrophy of cerebellar vermis and hemispheres in patient 1 at the age of 3 years (Fig 1A–1F). Progression of cerebellar atrophy and mild ventricular enlargement were observed in patient 1 at the age of 14 years (Fig 1G–1L). In patient 2 brain MRI at the age of 6 months was normal but showed the same abnormalities as in her brother at the age of 8 years (S1 Text and Table 1).

**Table 1. Clinical features of the siblings with the homozygous *CACNB4* mutation p.(Leu126Pro).**

| | Patient 1 | Patient 2 |
| --- | --- | --- |
| Ethnicity | Turkish | Turkish |
| Sex | Male | Female |
| Family history | Negative | Negative |
| Pregnancy | Uneventful | Uneventful |
| Birth at | Term | Term |
| **Measurements** | | |
| Birth weight (g/z) | 3840/0.7 | 3240/-0.2 |
| Birth length (cm/z) | 52/-0.1 | 52/0.4 |
| OFC at birth (cm/z) | 34/-1.1 | 34/-0.2 |
| Age at last examination | 15 years | 22 years |
| Weight at last examination (kg/z) | 30/-3.7 | 36.4/-4.1 |
| Height at last examination (cm/z) | 138/-3.8 | 151/-2.7 |
| OFC at last examination (cm/z) | 52.5/-2.1 | 53/-1.9 |
| **First clinical signs** | | |
| | Severe developmental delay, no eye contact | Severe developmental delay, no eye contact |
| **Neurological features** | | |
| Global developmental delay | ++ | ++ |
| Motor skills achieved<br> - rolling over<br> - sitting<br> - walking | <br>+<br>-<br>- | <br>-<br>-<br>- |
| Truncal muscular hypotonia | ++ | ++ |
| Spasticity | - | - |
| Athetoid-dystonic movements | + | + |
| Intellectual disability | ++ | ++ |
| Speech impairment | ++ | ++ |
| **Seizures** | | |
| Age of onset | 3 years | 6 months |
| Initial seizure type | Focal | Tonic |
| Current seizure type | Focal | Focal, tonic |
| EEG at last examination | Multiregional sharp waves, structural deficiency, slowed background activity | Multiregional sharp waves with secondary generalization, structural deficiency, slowed background activity |
| Response to treatment | Seizure-free on monotherapy | Intractable |
| **Other** | | |
| Feeding difficulties | + | + |
| Failure to thrive | + | + |
| Hearing | Normal | Normal |
| Eye contact | Reduced | None |
| Ophthalmologic examination (at age of) | No optic atrophy, normal retina (15 years) | No optic atrophy, normal retina (22 years) |
| Visual evoked potentials | Absent | Reduced |
| Cerebral MRI | Severe cerebellar atrophy, mild ventricular enlargement | Severe cerebellar atrophy, mild ventricular enlargement |

EEG: electroencephalogram; OFC: occipital frontal circumference, ++: severe, +: present, -: absent

Segregation analysis of the 12 rare homozygous variants in the two siblings and their parents excluded six to be associated with the patients' phenotype, but six variants remained

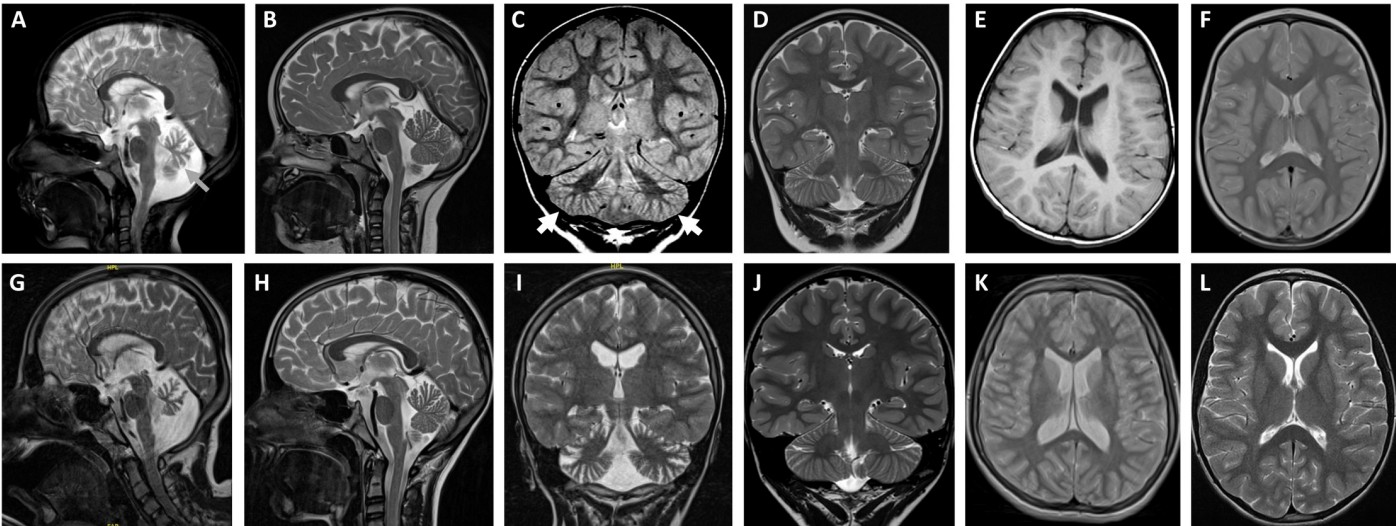

**Fig 1. Brain imaging of patient 1 revealed cerebellar atrophy.** (A, C and E) Selected brain MRI scans of patient 1 at the age of 3 years in comparison to age-matched normal scans (B, D and F; all T2-weighted). T2-weighted sagittal section (A) and coronal section (C) show moderate cerebellar atrophy including the vermis and hemispheres (indicated by arrows). (E) T1-weighted transversal section showing no supratentorial abnormalities. (G, I and K) Selected MRI scans of patient 1 at the age of 14 years in comparison to age-matched normal scans (H, J and L; all T2-weighted). T2-weighted sagittal section (G) and coronal section (I) demonstrate severe cerebellar atrophy. (K) Mild ventricular enlargement is observed on T2-weighted transversal section.

(S1 Table). Four of the six variants were absent in all population databases and affected the genes *LTBP1*, *EMILIN1*, *CACNB4*, and *MINAR1* (S1 Table). CADD, REVEL, and M-CAP, the pathogenicity prediction programs combining previous pathogenicity scores, inconsistently predicted the variants in *LTBP1*, *EMILIN1*, and *MINAR1* to have a damaging effect on protein function (S1 Table). The three genes have not yet been reported in the context of rare Mendelian disorders.

To further examine possible correlations of these genetic variants with the clinical features in the two patients, we checked the phenotype of the respective published mouse knockout model. Knockout of the long form of *Ltbp1* (*Ltbp1L*) in mice caused early postnatal lethality. *Ltbp1L*$^{-/-}$ mice had developmental abnormalities of the heart outflow tract, including persistent truncus arteriosus and interrupted aortic arch, and hypoplastic endocardial cushions. These data demonstrate an essential role of Ltbp1L during heart organogenesis and valvulogenesis [38, 39]. Recently, *LTBP1* has been reported as disease gene candidate for primary platelet secretion defects [40]. *Emilin1* knockout mice showed defects of elastic fibers in aorta and skin suggesting that Emilin1 is implicated in elastogenesis and maintenance of blood vascular cell morphology [41]. The identification of a heterozygous missense variant in *EMILIN1* in a proband with a connective disorder suggested this gene as a new disease gene for an autosomal-dominant connective tissue disorder [42]. To our knowledge, a mouse model for *Minar1* (alternative names *KIAA1024* and *UBTOR*) does not yet exist. First functional data suggest a role of MINAR1 in angiogenesis [43]. Recent data indicate UBTOR/KIAA1024 to regulate cellular growth and mTOR signaling. Homozygous *ubtor* zebrafish mutants had no gross developmental abnormalities. Behavioral tests showed enhanced fear-evoked freezing and compromised C-start responses in mutant fish, suggesting a possible role of ubtor in neurodevelopment [44]. However, *KIAA1024/MINAR1/UBTOR* has not been reported as candidate gene for intellectual disability in large whole-exome sequencing studies [45–49]. Implication of Ltbp1 in heart development [50] and Emilin1 in skin homeostasis and blood pressure control [51] do not support any contribution of the variants in *LTBP1* and *EMILIN1* to the

neurological anomalies in the two siblings. However, an effect of the *MINAR1* variant p.(Ser855Tyr) on the patients' phenotype cannot yet be excluded based on limited data from literature.

Two missense variants with a MAF of 0.0016% and 0.0012% were identified in known disease genes (S1 Table): c.889C>T/p.(Arg297Cys) in *DHX38* (MIM: 605584) in which two other amino acid substitutions have been reported in individuals with autosomal recessively inherited early-onset retinitis pigmentosa (MIM: 268000) [52, 53] and c.5083C>A/p.(Pro1695Thr) in *MYO15A* (MIM: 602666) in which biallelic mutations cause autosomal-recessive, nonsyndromic deafness (DFNB3, MIM: 600316) [54, 55]. The *MYO15A* change c.5083C>A is no known disease-associated allele. Although three *in silico* tools predicted the variant to be pathogenic (S1 Table), neither one of the siblings had any hearing problems (S1 Text and Table 1), suggesting that this nucleotide change likely represents a rare polymorphism rather than a pathogenic mutation. The *DHX38* variant p.(Arg297Cys) was predicted to be possibly damaging by two of three programs (S1 Table). In individuals with a pathogenic *DHX38* variant, blindness is caused by retinitis pigmentosa and occurred between 7 and 8 years of age, and the majority of affected individuals developed cataract by the age of 19 years [52]. In contrast, the diagnosis of blindness in patients 1 and 2 described here was established within the first year of their life, and ophthalmologic examination at age 15 years and 22 years, respectively, did not reveal any signs of cataract and/or retinitis pigmentosa. Most likely cortical blindness accounted for visual loss in both individuals reported here (S1 Text). Together, these data suggest that the *DHX38* c.889C>T change is a benign variant and not causative for visual impairment in the two siblings.

The homozygous variant c.377T>C/p.(Leu126Pro) in the disease-associated gene *CACNB4* was predicted to be pathogenic by all three *in silico* tools with exceptional high scores (S1 Table). Heterozygous *CACNB4* variants have been implicated in epilepsy and episodic ataxia [15], and the *Cacnb4* knockout mice have a severe neurological phenotype [19, 21, 56]. The p.(Leu126Pro) amino acid change affects a highly conserved leucine in the SH3 domain of β4. In fact, equivalent substitutions in β$_{1a}$ and β$_{2a}$ disrupt the functionally critical tandem SH3-GK module of Ca$_V$ β subunits, similarly to that in MAGUK proteins [31, 57–59]. To obtain more evidence for a possible disease association of the *CACNB4* variant p.(Leu126Pro), also in light of the other five homozygous variants identified in the two affected siblings, we first queried available genomic resources and large genome-wide sequencing studies for additional individuals with biallelic *CACNB4* variants, including DECIPHER and the DDD study. We did not identify any homozygous or compound heterozygous variants in this gene [46, 47, 49, 60–68]. Through GeneMatcher [69] we did not get a match reporting biallelic variants in *CACNB4*. The absence of further individuals with biallelic variants in this gene suggests the presence of an ultra-rare genetic disease in the two affected siblings.

We next compared the constraint score of the observed/expected (o/e) number of missense variants for the genes *LTBP1*, *EMILIN1*, *CACNB4*, *MINAR1*, *DHX38* and *MYO15* in gnomAD. The o/e score measures the tolerance of a gene to a certain class of variation. A low o/e value indicates that the gene is under stronger selection than a gene with a higher score. Among the six genes, *CACNB4* has the lowest o/e score for missense variants (0.55) indicating that this gene is under selection and probably a Mendelian disease gene (S1 Table). In addition, we used the MetaDome web server, which provides profiles of genetic tolerance through aggregation of homologous human protein domains [70]. MetaDome predicted leucine 126 of CACNB4 to be highly intolerant, while the genetic tolerance of the amino acid residues affected in the other five genes ranges from neutral to intolerant (S1 Table). Together, by a combination of *in silico* tools, animal model, clinical, and genetic data, we suggest that the

homozygous p.(Leu126Pro) amino acid substitution in *CACNB4* is the likely variant to underlie the patients' neurological disease, although there is still a degree of uncertainty.

## The analogous *Cacnb4* mutation in rat, p.Leu125Pro, impairs the association of $\beta_{4b}$ with calcium channel complexes and $\beta_{4b}$ nuclear targeting in muscle cells and neurons

In order to examine possible functional defects of the *CACNB4* p.Leu126Pro mutation that might explain the neurological phenotype in the patients, we studied the effects of the amino acid substitution in muscle and nerve cells, two well-established and differentiated cellular expression systems for voltage-gated calcium channels [27, 28, 71]. Based on the functional defects described for an equivalent substitution in $\beta_{2a}$ [31], we hypothesized that the corresponding substitution of leucine 125 by proline in rat $\beta_{4b}$ might hamper its association with pore-forming $Ca_V$ subunits. To test this, we first expressed V5-tagged $\beta_{4b}$, $\beta_{4b}$-L125P, and $\beta_{1a}$ together with the L-type calcium channel $Ca_V1.2$ in dysgenic ($Ca_V1.1$-null) myotubes. This expression system has been extensively used to study the structural and functional incorporation of calcium channels and associated proteins in a native signaling complex of a differentiated excitable cell [72–76]. Fig 2A (left panel) shows that GFP-$Ca_V1.2$ and the skeletal muscle $\beta_{1a}$ isoform, labeled with anti-GFP and anti-V5, respectively, co-localize in clusters corresponding to sarcoplasmic reticulum (SR)/plasma membrane and SR/T-tubule junctions, collectively referred to as triads. Because $\beta$ subunits require binding to a $Ca_V1$ subunit for their incorporation in triads, this co-clustering is indicative of a stable $Ca_V1.2/\beta_{1a}$ interaction [74, 77]. Similarly to the skeletal muscle $\beta_{1a}$ isoform and in line with previous findings [28, 71], the wild-type neuronal $\beta_{4b}$ isoform co-clustered with $Ca_V1.2$ in the triads (Fig 2A, center panel), showing that it too associates with the channel. However, the mutant $\beta_{4b}$-L125P failed to associate with $Ca_V1.2$ (Fig 2A, right panel). In contrast to $\beta_{1a}$ and $\beta_{4b}$, no transfected myotubes showed co-clustering of $\beta_{4b}$-L125P with $Ca_V1.2$ in triads. Instead, $\beta_{4b}$-L125P remained diffusely distributed in the cytoplasm, while $Ca_V1.2$ was clustered in triads (Fig 2B). Thus, the p. Leu125Pro substitution disrupts the association of $\beta_{4b}$ with the $Ca_V1.2$ channel complex in myotubes.

In addition to the clustered triad labeling, myotubes transfected with wildtype $\beta_{4b}$ displayed strong nuclear staining (Fig 2A). A similar nuclear localization was not observed with $\beta_{1a}$ and thus represents isoform-specific nuclear targeting of $\beta_{4b}$ that has previously been demonstrated [28]. Importantly, mutated $\beta_{4b}$-L125P was excluded from the nuclei (Fig 2A). Quantitative analysis showed that 52.5±4.4% of the cells transfected with $\beta_{4b}$ displayed nuclear targeting, whereas no cells with nuclear staining were found for $\beta_{4b}$-L125P (Fig 2C). Note that $\beta_{4b}$ nuclear targeting is negatively regulated by electrical activity [26, 28], and therefore a subset of, probably spontaneously active, myotubes displayed no nuclear targeting, even when transfected with wildtype $\beta_{4b}$ (Fig 2C). Compromised nuclear targeting of $\beta_{4b}$-L125P was further corroborated by the significant difference of the nucleus/cytoplasm ratio of staining intensities (Fig 2D). Together, the co-expression experiments in the myotube model system indicate that the p.Leu125Pro mutation disrupts both, the association of $\beta_{4b}$ with the channel complex in the triad as well as $\beta_{4b}$'s nuclear targeting property.

Because the parents of the two affected siblings, who are heterozygous carriers of the p. (Leu126Pro) variant, were healthy, we examined channel association and nuclear targeting of wild-type ($\beta_{4b}$-GFP) and mutant $\beta_{4b}$ (L125P-V5) co-expressed together with Cav1.2 in dysgenic myotubes. As expected wild-type $\beta_{4b}$-GFP displayed a clustered staining pattern, consistent with its normal incorporation into channel complexes. In contrast, in the same cells $\beta_{4b}$-L125P-V5 remained diffusely distributed in the cytoplasm. Similarly, in cells that showed $\beta_{4b}$-

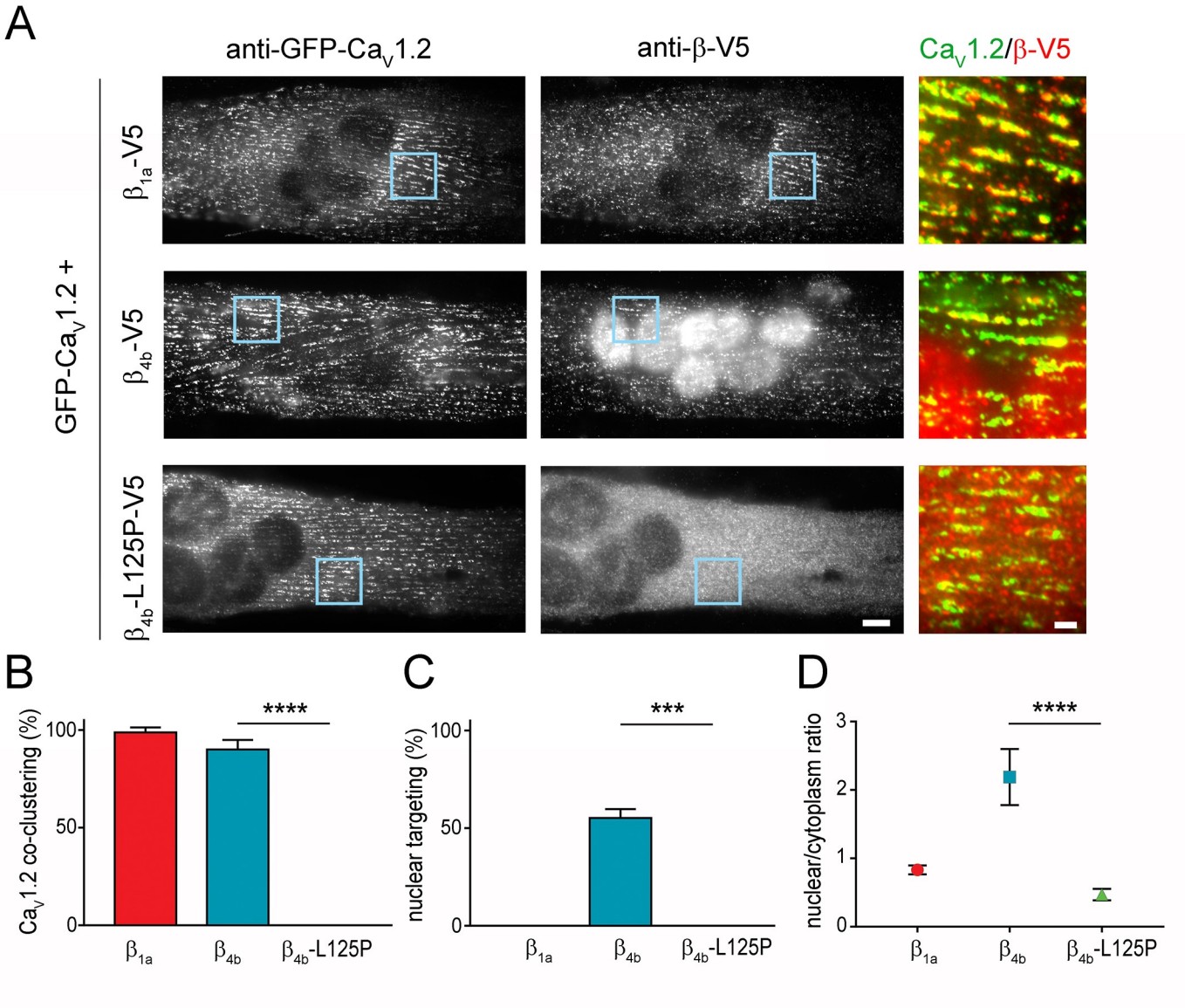

**Fig 2. The p.Leu125Pro mutation impairs co-clustering with Ca$_V$1.2 and nuclear targeting of β$_{4b}$ in skeletal myotubes.** (A) Representative double-immunofluorescence images of myotubes transfected with β$_{1a}$-V5, β$_{4b}$-V5, or β$_{4b}$-L125P-V5 expression construct together with GFP-Ca$_V$1.2, labeled with anti-GFP and anti-V5. Scale bar: 10 μm. Color overlay of GFP-Ca$_V$1.2 (green) and Ca$_V$β$_{1a}$ (red) staining; 4× magnification of regions indicated by blue rectangle. Scale bar, 2 μm. GFP-Ca$_V$1.2 was incorporated in triads (clusters) and both wildtype β$_{1a}$ and β$_{4b}$ subunits co-assembled with these calcium channel complexes. In addition, wildtype β$_{4b}$ specifically accumulated in the nuclei of the myotubes. In contrast, β$_{4b}$-L125P failed to co-cluster with Ca$_V$1.2 in triads and failed to target into the nuclei. (B) Fraction of myotubes in which the transfected β subunit co-clustered with GFP-Ca$_V$1.2 (N = 4; n = 240). ANOVA: F(2,9) = 1182, $P < 0.0001$. (C) Fraction of myotubes showing nuclear targeting (N = 4; n = 240). ANOVA: F(2,6) = 41.25, $P = 0.0003$. (D) Nucleus/cytoplasm ratios of myotubes labeled with anti-V5 (N = 3; n = 60). ANOVA: F(2,177) = 47.75, $P < 0.0001$.

GFP nuclear targeting, mutant β$_{4b}$-L125P-V5 were excluded from the nuclei (S1 Fig). These results demonstrate that in the presence of mutant β$_{4b}$-L125P the wild-type β$_{4b}$ subunit still displays its normal association with calcium channel complexes as well as its nuclear targeting properties.

Next, we sought to determine whether the p.Leu125Pro mutation also affects the targeting properties of β$_{4b}$ in neurons. Previously, we demonstrated that β$_{4b}$ is strongly expressed in the

somatodendritic compartment as well as the proximal and distal axon of cultured hippocampal neurons [27]. This was most evident when co-expressing V5 epitope-tagged $\beta_{4b}$ together with soluble eGFP in order to outline the dendritic and axonal branching pattern and imaging the arborization of individual transfected neurons (Fig 3A, left panel). It is important to note that blacklevel and contrast of these overview images were adjusted in a way to visualize also the weak staining in the distal axon. Hence, potential staining differences in the somata (cf. Fig 4) are not visible at these settings. $\beta_{4b}$ immunostaining was strong throughout the soma and the dendrites as well as the proximal parts of the axon (Fig 3A, left panel, arrows) and, importantly, $\beta_{4b}$ labeling was also found in the distal and fine axonal branches. Quantitative analysis of anti-V5 labeling intensity in the distal axon revealed a significant reduction of $\beta_{4b}$-L125P-V5 compared with $\beta_{4b}$-V5 (fluorescence intensity above background: 1.33±0.07 [$\beta_{4b}$-V5, n = 14] and 1.03±0.05 [$\beta_{4b}$-L125P-V5, n = 18], $t_{(30)}$ = 4.42, p<0.001). Where an axon contacted neighboring, non-transfected cells, this pattern appeared dotted, which is typical for an accumulation of $\beta_{4b}$ in presynaptic boutons (Fig 3A, left panel, arrowheads). In contrast, expression of the $\beta_{4b}$-L125P mutant was largely restricted to the soma and dendrites (Fig 3A, right panel). Moreover, the proximal axon segments were only faintly labeled (Fig 3A, right panel, arrows) and a dotted pattern at axonal contact points with neighboring cells was entirely missing (Fig 3A, right panel, arrowheads).

$\beta_{4b}$ co-localizes with the P/Q-type calcium channel Ca$_V$2.1 in presynaptic nerve terminals of differentiated cultured hippocampal neurons [27]. Because our overview imaging suggested little to no axonal targeting of $\beta_{4b}$-L125P (Fig 3A), we next performed high-resolution imaging of presynaptic boutons in order to test whether any detectable synaptic localization of the $\beta_{4b}$ mutant was retained (Fig 3B). To this end, we followed the eGFP-positive axon of transfected neurons to distal contact points with untransfected neighboring neurons. Whenever such axons contact non-transfected dendrites or cell somata they form axonal varicosities, which are typical hallmarks of presynaptic nerve terminals [27]. Such axonal varicosities were clearly visible after expression of diffusible eGFP in neurons (e.g. Fig 3B, eGFP). Wild-type $\beta_{4b}$, which was abundant throughout the axons (see above), specifically localized in clusters coinciding with varicosities of the eGFP-labeled axons (Fig 3B, left panel, $\beta_{4b}$ + eGFP, examples indicated by arrowheads). Most importantly, $\beta_{4b}$-L125P did not accumulate in presynaptic nerve terminals as identified by eGFP-positive varicosities lacking any corresponding anti-V5 staining (Fig 3B, right panel, examples indicated by arrowheads). Together, these observations indicate that the $\beta_{4b}$-L125P mutant fails to be trafficked into the axon and to cluster in synaptic terminals in hippocampal neurons, suggesting that it cannot be incorporated into presynaptic calcium channel complexes.

To analyze the nuclear targeting properties of $\beta_{4b}$ and $\beta_{4b}$-L125P in neurons, we focused our attention on the somata of hippocampal neurons (Fig 4). In untreated cultures, wild-type $\beta_{4b}$ showed a uniform pattern, with labeling of the subunit in both the cytoplasm and the nucleus (Fig 4A, left panel, anti-V5). In contrast, mutant $\beta_{4b}$-L125P was excluded from the nuclei (Fig 4A, right panel, anti-V5). This difference was most evident when comparing the nucleus/cytoplasm ratio of the anti-V5 labeling intensity between $\beta_{4b}$ (0.99±0.16, mean±SD) and $\beta_{4b}$-L125P (0.76±0.11, mean±SD), which was significantly different (Fig 4B). Previously, we demonstrated that $\beta_{4b}$ nuclear targeting is negatively regulated by electrical activity [26, 28]. Therefore, we blocked spontaneous neuronal activity by overnight application of a 1 µM concentration of the sodium channel blocker tetrodotoxin (TTX). As expected, in TTX-treated hippocampal neurons $\beta_{4b}$ strongly accumulated in the nuclei (Fig 4C, left panel, anti-V5), and the mean nucleus/cytoplasm ratio increased to 1.46±0.30 (mean±SD, compare Fig 4B and 4D). Nonetheless, $\beta_{4b}$-L125P remained entirely cytoplasmic in TTX-treated neurons (Fig 4C, right panel, anti-V5) and consequentially the mean nucleus/cytoplasm ratio (0.79±0.16, mean±SD,

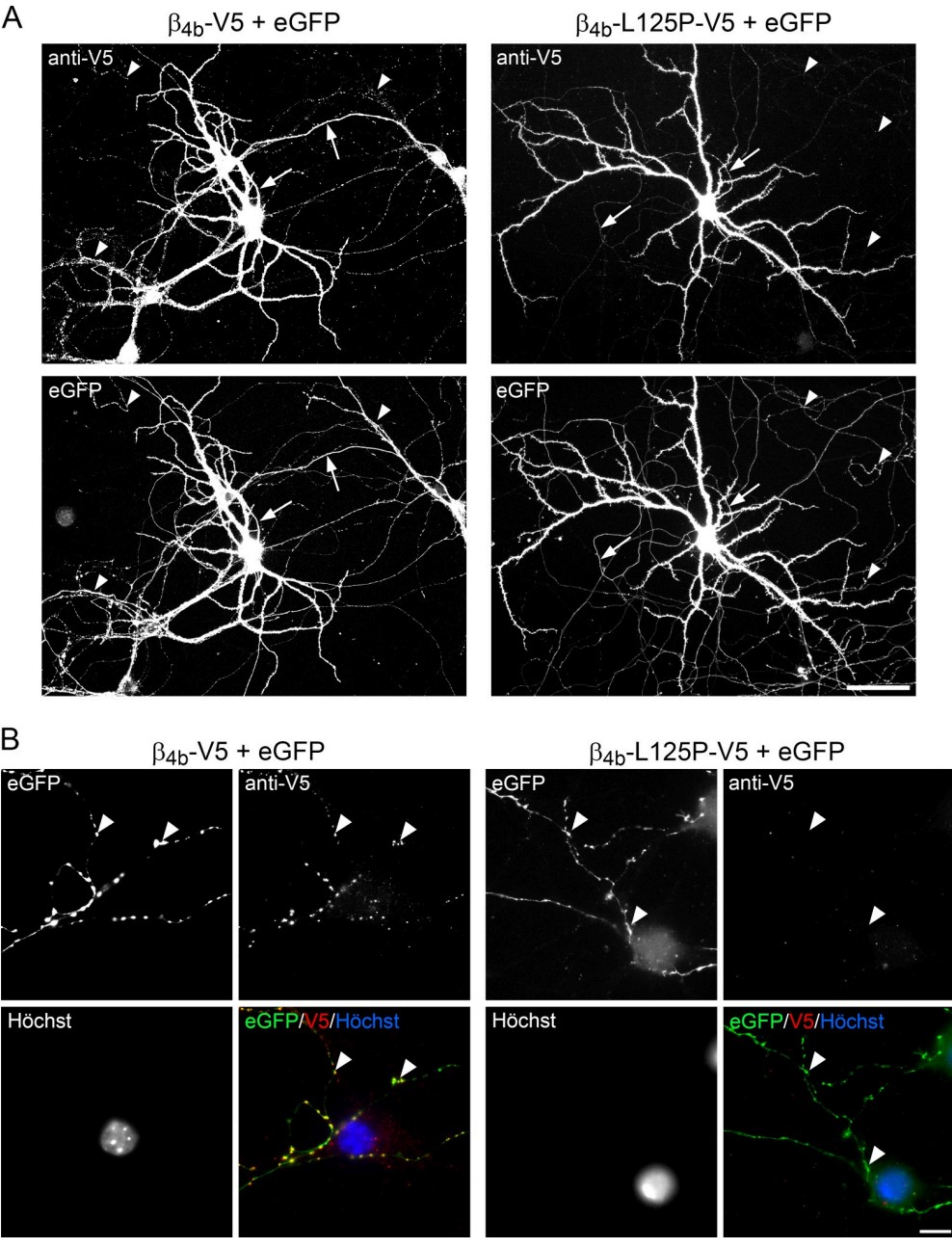

**Fig 3. Axonal and synaptic targeting of the β4b-L125P mutant fails in primary hippocampal neurons.** Cultured hippocampal neurons were transfected with eGFP and β4b-V5 or β4b-L125P-V5 on DIV6 and immunolabled with anti-V5 on DIV34. Nuclei were labeled with Hoechst dye (blue). (A) Overview images of the dendritic and axonal arborization (eGFP) and anti-V5 labeling (anti-V5) of neurons expressing β4b-V5 (left panel) or β4b-L125P-V5 (right panel). Arrows indicate segments of the proximal axons and arrowheads indicate exemplary axonal contact points with neighboring non-transfected cells typical for presynaptic boutons. (B) Presynaptic boutons were identified as eGFP-filled axonal varicosities contacting postsynaptic untransfected cell somata or dendrites. β4b-V5 accumulated in presynaptic boutons of transfected hippocampal neurons (arrowheads, left panel). In contrast, a distal axonal localization of β4b-L125P-V5 was missing and β4b-L125P-V5 failed to be incorporated in presynaptic boutons (arrowheads, right panel). Scale bars, 20 μm (A) and 10 μm (B). Representative images of three independent experiments are shown (see Results for statistics).

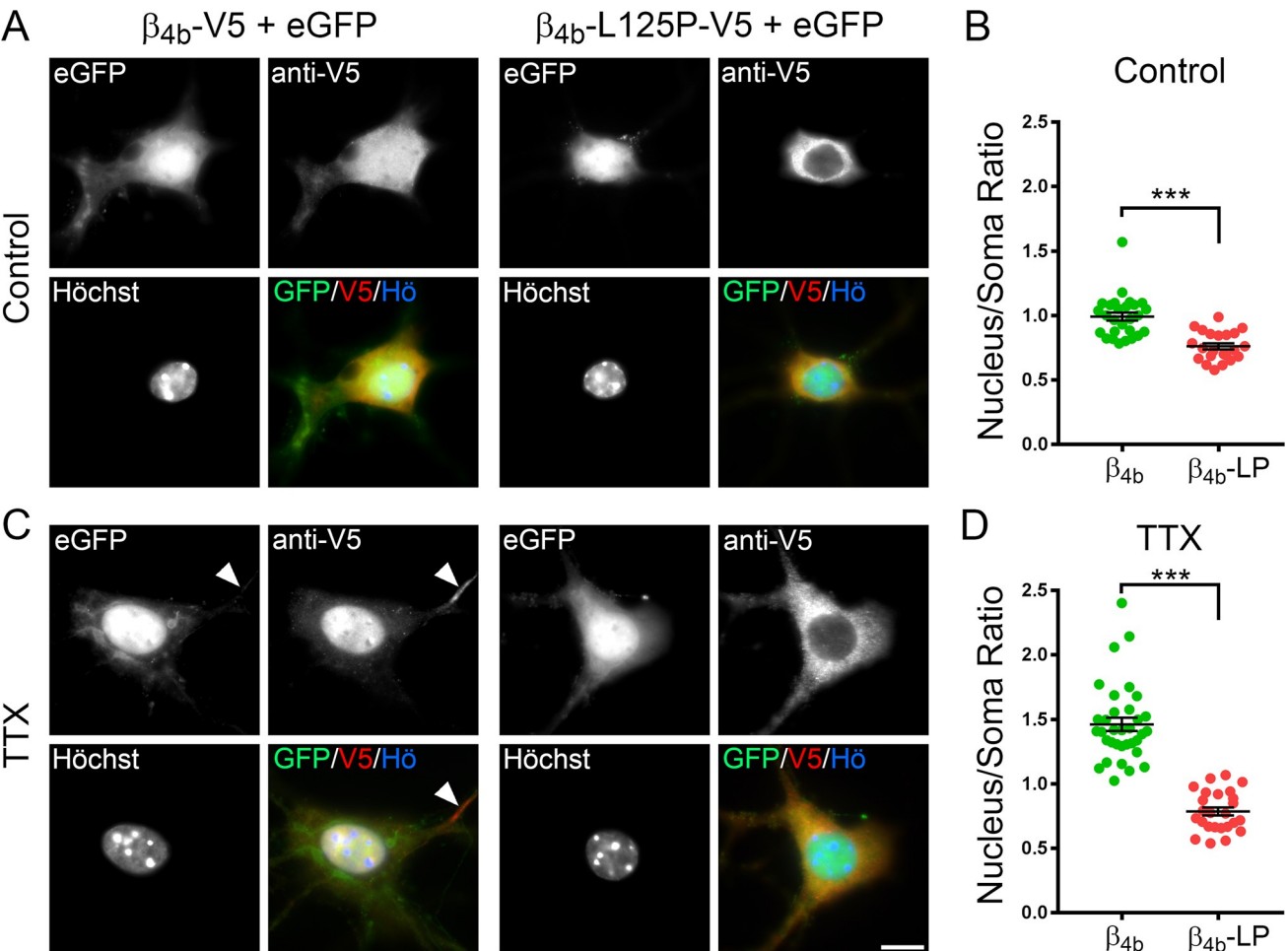

**Fig 4. The p.Leu125Pro mutation impairs activity-regulated nuclear targeting of β₄ᵦ in hippocampal neurons.** Cultured hippocampal neurons were transfected with eGFP and β₄ᵦ-V5 or β₄ᵦ-L125P-V5 on DIV6 and fixed and fluorescently labeled with anti-V5 (red) and Hoechst dye (blue) on DIV34 (control) or DIV35 (TTX). TTX treatment (1μM) was performed overnight (12h). (A) In untreated neurons (control) β₄ᵦ-V5 was distributed throughout the soma, dendrites and axons (not visualized) and, to a low degree, also in the nucleus. In contrast, localization of β₄ᵦ-L125P-V5 was more strongly restricted to the cell soma and largely excluded from the nucleus. (B) Nucleus/cytoplasm ratio of anti-V5 labeling intensity. (C) Overnight silencing of electrical activity with TTX induced a strong accumulation of β₄ᵦ-V5 in the cell nuclei while β₄ᵦ-L125P-V5 failed to localize to the nucleus. Note that wild-type β₄ᵦ-V5 (but not β₄ᵦ-L125P-V5) also accumulated in the axon hillock (arrowhead, see Results for statistics). (D) Nucleus/cytoplasm ratio of β₄ᵦ-V5 and β₄ᵦ-L125P-V5. Statistics (B and D), 2-way ANOVA: Transfection: $F_{(1, 104)} = 166$, $P < 0.001$; treatment (control, TTX): $F_{(1, 104)} = 35$, $P < 0.001$; transfection x treatment: $F_{(1, 104)} = 27$, $P < 0.001$; Holm-Sidak *posthoc* analyses showed $P < 0.001$ (***) in all pairwise comparisons except for treatment (TTX, control) within β₄ᵦ-L125P ($P = 0.64$). A total number of 108 cells (between 22 and 34 in each treatment group) from two separate culture preparations was analyzed. Scale bar, 10 μm.

Fig 4D) was indistinguishable from that of untreated control neurons (compare Fig 4B and 4D). These data indicate that the p.Leu125Pro mutation abolishes the basal and activity-dependent nuclear localization of β₄ᵦ in neurons. Finally, wild-type β₄ᵦ-expressing neurons showed a specific localization of β₄ᵦ in the axon hillock (Fig 4C, left panel, anti-V5, arrowhead), as previously reported [27]. This distinctive localization was not observed in β₄ᵦ-L125P-expressing neurons (fluorescence intensity above background: 5.07±0.66 [β₄ᵦ-V5, n = 27] and 1.92±0.17 [β₄ᵦ-L125P-V5, n = 21], $t_{(46)} = 4.15$, p<0.001). Altogether, these results demonstrate that substitution of leucine 125 by proline inhibits (1) axonal trafficking and synaptic localization, (2) basal and activity-dependent nuclear targeting, and (3) the specific β₄ᵦ accumulation in the axon hillock of cultured hippocampal neurons. This is likely caused by disrupting (1) the

interaction of $\beta_{4b}$ with presynaptic calcium channel complexes, (2) the machinery responsible for $\beta_{4b}$ accumulation in the nuclei of quiescent neurons, and (3) the interaction with proteins of unknown nature in the axon hillock.

### The p.Leu125Pro mutation does not abrogate complex formation of $\beta_{4b}$ with PPP2R5D (B56δ), but with the TRAF2 and NCK interacting kinase (TNIK)

Rat $\beta_4$ has previously been shown to form a protein complex with Ppp2r5d (B56δ), a regulatory subunit of protein phosphatase 2A, that contributes to nuclear localization of $\beta_4$. Consistent with previous findings [29] we demonstrated that substitution of leucine 125 by proline impairs nuclear targeting of $\beta_{4b}$ in cultured hippocampal neurons (Fig 4). To examine the consequence of the p.Leu125Pro mutation on the interaction of $\beta_{4b}$ with PPP2R5D, we immunoprecipitated endogenous PPP2R5D from HEK293T cells transfected with V5-tagged $\beta_{4b}$ wildtype or $\beta_{4b}$-L125P and detected $\beta_{4b}$ in the precipitates by immunoblotting using anti-V5 or anti-$\beta_4$ antibodies. V5-tagged $\beta_{1a}$ was used as negative control as this protein could not be co-precipitated with PPP2R5D (Fig 5A). In contrast, both wild-type $\beta_{4b}$ and the $\beta_{4b}$-L125P mutant co-precipitated with endogenous PPP2R5D, as low amounts of $\beta_{4b}$ wildtype and mutant were detected in the immunoprecipitates by anti-V5 and anti-$\beta_4$ antibodies (Fig 5A, right panel). These data demonstrate that the p.Leu125Pro mutation did not abrogate the modest complex formation between $\beta_{4b}$ and PPP2R5D in HEK293T cells.

Next, we examined whether the $\beta_{4b}$-L125P mutant alters subcellular localization of endogenous PPP2R5D. Previous data suggested that expression of eGFP-tagged β4 in CHO-K1 cells causes an enrichment of PPP2R5D in the nucleoplasm [78]. We ectopically expressed V5-tagged wild-type $\beta_{4b}$ and $\beta_{4b}$-L125P mutant in HEK293T cells and stained the cells for endogenous PPP2R5D and $\beta_{4b}$ using anti-PPP2R5D and anti-V5 antibodies, respectively. In HEK293T cells transfected with empty vector, PPP2R5D was diffusely located in the cytoplasm, but also showed some nuclear staining (Fig 6). As shown in myotubes (Fig 2) and primary hippocampal neurons before (Fig 4), V5-tagged $\beta_{4b}$ wildtype was also enriched in the nuclei of HEK293T cells (Fig 6). In contrast, $\beta_{4b}$-L125P transfected cells exhibited a predominant cytoplasmic distribution, and $\beta_{4b}$-L125P was excluded from the nucleus. Importantly, the cellular distribution of PPP2R5D did not change upon expression of $\beta_{4b}$ wildtype or $\beta_{4b}$-L125P in HEK293T cells. Neither did we observe increased nuclear PPP2R5D staining with co-expressed $\beta_{4b}$ compared to control (empty vector), nor any change with $\beta_{4b}$-L125P (Fig 6). Together, the data suggest that complex formation between PPP2R5D and $\beta_{4b}$ may not be necessary or sufficient for nuclear targeting of one or the other protein.

In search of alternative neuronal $\beta_{4b}$ binding partners, we screened the BioGRID database (https://thebiogrid.org) [79] for CACNB4/$\beta_4$ and identified the TRAF2 and NCK interacting kinase (TNIK) [80]. TNIK is expressed in the brain and has been suggested to be important for postsynaptic signaling, neurogenesis and cell proliferation [81–83]. We found endogenous TNIK to be expressed in HEK293T cells (Fig 5B, left panel) and then aimed to study if TNIK is in complex with $\beta_{4b}$. We expressed V5-tagged $\beta_{4b}$ wildtype in HEK293T cells, immunoprecipitated endogenous TNIK and detected $\beta_{4b}$ in the precipitates. $\beta_{4b}$ wildtype was efficiently co-precipitated with TNIK (Fig 5B, right panel). In contrast to wild-type $\beta_{4b}$, we consistently failed to detect the $\beta_{4b}$-L125P mutant in the immunoprecipitates by using anti-V5 or anti-$\beta_4$ antibody (Fig 5B, right panel). These data demonstrate that the p.Leu125Pro mutation abrogates the capability of $\beta_{4b}$ to form a protein complex with neuronal TNIK.

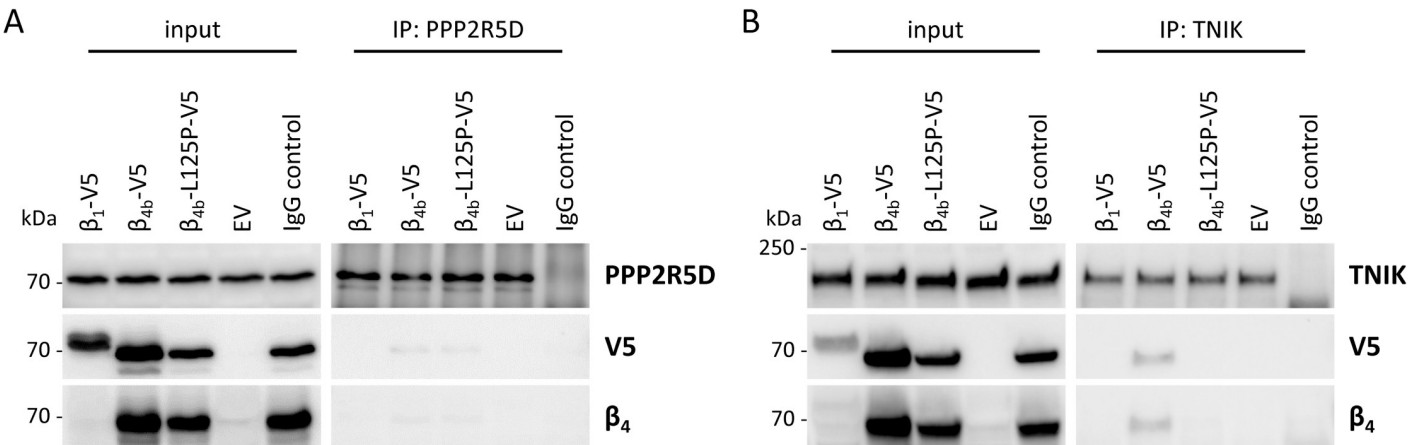

**Fig 5. The p.Leu125Pro mutation does not disrupt complex formation of β4b with PPP2R5D, but with TNIK.** (A and B) HEK293T cells were transfected with empty vector (EV), β1-V5, β4b-V5 or β4b-L125P-V5 expression construct as indicated. Endogenous PPP2R5D and TNIK were immunoprecipitated from cell extracts using an anti-PPP2R5D (A) and an anti-TNIK antibody (B), respectively, both coupled to magnetic protein G Dynabeads. For IgG control the cell lysate from cells transfected with the β4b-L125P-V5 mutant construct was incubated with an anti-normal rabbit IgG antibody coupled to Dynabeads. Co-precipitated β1-V5 and β4b-V5 proteins were detected by immunoblotting using anti-V5-HRP and anti-β4 antibody. A representative blot of four (A) or three (B) independent experiments each is shown.

## The p.Leu125Pro mutation does not abolish the augmentation of calcium currents by β4b

The observed loss of β4b's stable association with calcium channel complexes in synapses and triads suggested that the ability of the β4b-L125P mutant to augment calcium currents may also be compromised. To examine this possibility, we heterologously co-expressed wild-type and mutant β4b subunits with GFP-tagged Ca$_V$2.1 plus the α2δ-1 subunit in tsA201 cells and analyzed whole cell barium currents using patch-clamp electrophysiology (Fig 7). When Ca$_V$2.1 and α2δ-1 were expressed without a β subunit in tsA201 cells, currents were below detectability in about 70% of the cells, and the analyzable cells showed currents of low amplitude (3.32 ±1.15pA/pF; mean ±SEM) (Fig 7A and 7B). Upon co-expression of wildtype β4b the peak current density increased >10-fold (36.79 ±7.25 pA/pF) (Fig 7A and 7B), consistent with the known function of β subunits in increasing functional expression of calcium channels in the membrane of heterologous cells [9, 84]. This significant increase in current amplitude was accompanied by a modest, but highly significant reduction of current inactivation. The residual fractional current at the end of the 200 ms test pulse increased from 48% to 77% (Fig 7E and 7F). Voltage-dependence of activation was not altered by co-expression of β4b (Fig 7C and 7D and S2 Table). Surprisingly, co-expression of the mutant β4b-L125P also caused a significant increase in current density (35.23 ±6.38 pA/pF), similar to that of the wildtype β4b (Fig 7A and 7B). Notably, however, the β effect on current inactivation was less pronounced with β4b-L125P compared to β4b. The mean size of the residual currents after 200 ms in cells co-expressing Ca$_V$2.1 with β4b-L125P (64%) was between the values recorded in cells expressing Ca$_V$2.1 with β4b and those without β, and this difference was significant relative to both conditions (Fig 7E and 7F). Thus, the β4b-L125P mutant retains the ability to augment Ca$_V$2.1 currents in tsA201 cells, whereas its effects on Ca$_V$2.1 current inactivation are slightly reduced by the single residue substitution.

## Heterologously expressed P/Q-type Ca$_V$2.1 and L-type Ca$_V$1.2 channels do not efficiently co-precipitate with β4b-L125P

The severe effects on channel association and synaptic targeting in native cell systems (Figs 2 and 3) are only partially mirrored by defects in channel modulation and not at all in channel

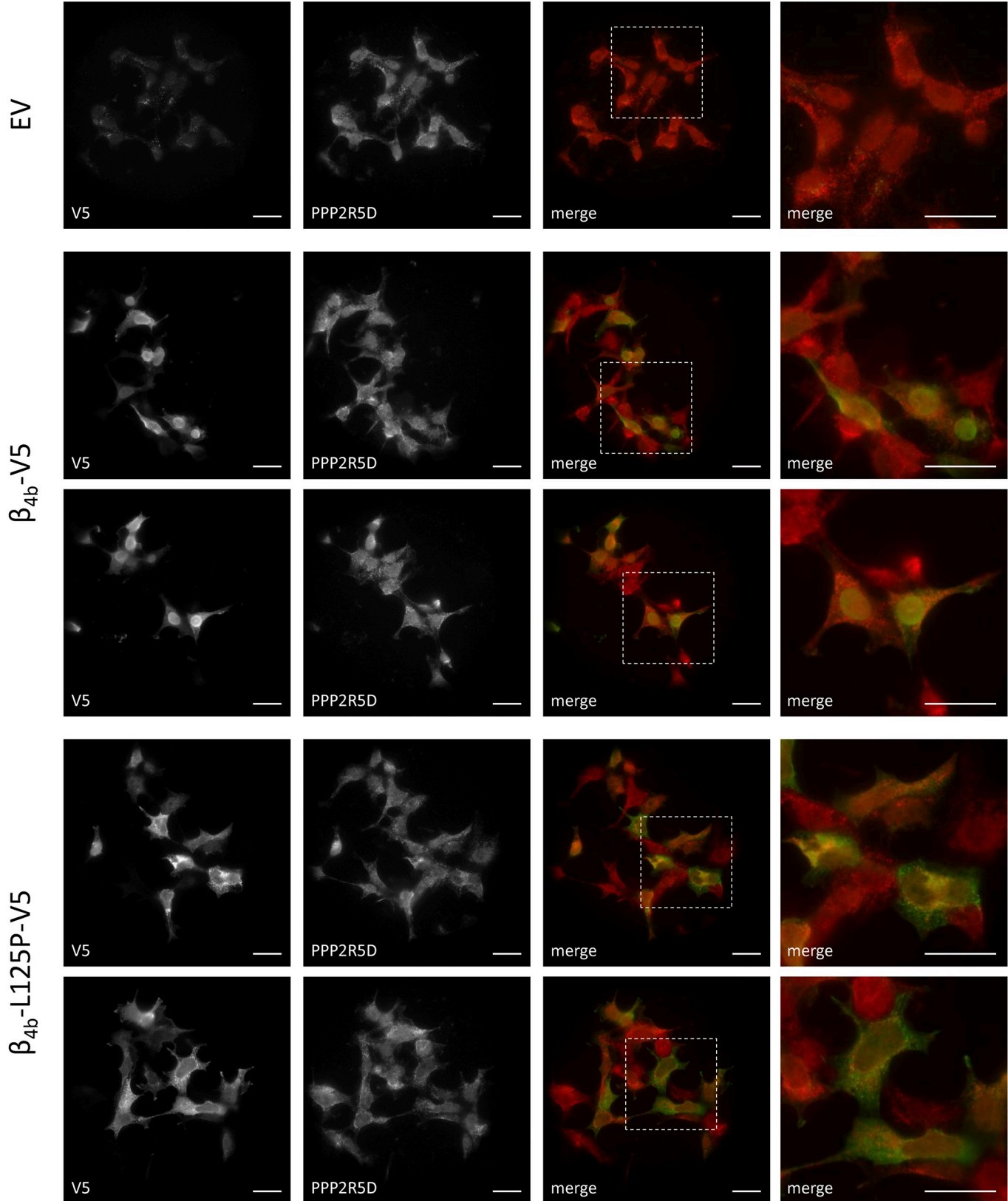

**Fig 6. Subcellular localization of PPP2R5D is not affected by expression of β$_{4b}$ wildtype or the L125P mutant in HEK293T cells.** HEK293T cells were plated on collagen-coated glass slides and transiently transfected with the indicated constructs. β$_{4b}$-V5 and β$_{4b}$-L125P-V5 were stained by mouse anti-V5 antibody (green); endogenous PPP2R5D was visualized using rabbit anti-PPP2R5D antibody (red). Representative images of two independent experiments are shown. White boxes indicate magnified areas of specimen shown on the very right-hand side. Scale bars, 10 μm. EV: empty vector (control).

membrane trafficking in heterologous cells (Fig 7). Previously, it has been shown that a decrease in the affinity of the β/α$_1$ subunit interaction affects calcium channel complex formation but not current modulation, the latter only requiring a high local concentration of β subunits [85–87]. Hence, we hypothesized that functional membrane expression of Ca$_V$ α$_1$ subunits in tsA201 cells may be achieved by high ectopic expression of β$_{4b}$-L125P which binds Ca$_V$2.1 with reduced affinity. In contrast, correct subcellular targeting and incorporation in native calcium channel complexes of differentiated nerve cells require a stable subunit interaction of wildtype β$_{4b}$ with pore-forming Ca$_V$ α$_1$ subunits. To test this possibility, we co-expressed wildtype β$_{4b}$ or β$_{4b}$-L125P in HEK293T cells together with the auxiliary α2δ-1 subunit and the P/Q-type Ca$_V$2.1 or L-type Ca$_V$1.2 channel α$_1$ subunit and immunoprecipitated the V5-tagged β$_{4b}$ subunit (Fig 8). As expected, both Ca$_V$2.1 and Ca$_V$1.2 were co-precipitated with wild-type β$_{4b}$ (Fig 8, left panel). In contrast, co-immunoprecipitation of Ca$_V$2.1 and Ca$_V$1.2 with the β$_{4b}$-L125P mutant was drastically decreased (Fig 8), indicating reduced complex formation between the β$_{4b}$-L125P mutant and the two α$_1$ subunits. Together, the functional, structural, and biochemical data show that although β$_{4b}$-L125P can still modulate the currents of heterologously expressed Ca$_V$2.1 channels, the mutation prevents formation of a stable complex between β$_{4b}$ and the α$_1$ subunit.

## Discussion

We report two siblings with severe intellectual disability lacking any language and motor development, seizures, visual impairment, and movement disorder who carry the homozygous p.(Leu126Pro) mutation in *CACNB4* encoding the cytoplasmic β$_4$ subunit of P/Q-type calcium channels. Functional analysis of the corresponding mutation in rat β$_{4b}$ (L125P) in a range of cellular model systems provides several lines of evidence supporting the notion that this mutation underlies the severe neurological phenotype in the patients: (1) The mutant β$_{4b}$-L125P completely failed to associate with native calcium channel complexes in the cultured myotube model system; (2) in hippocampal neurons axonal targeting of β$_{4b}$-L125P was severely compromised, resulting in the lack of β$_{4b}$-L125P in presynaptic boutons; (3) the mutation completely abolished basal and activity-dependent nuclear targeting of β$_{4b}$-L125P; (4) complex formation with the novel β$_{4b}$ interaction partner TNIK was disrupted in β$_{4b}$-L125P; (5) when co-expressed with Ca$_V$2.1 in tsA201 cells β$_{4b}$-L125P still supported functional membrane expression of the channels, however, complex formation with high-voltage activated calcium channels was strongly reduced.

### The homozygous *CACNB4* mutation p.Leu126Pro causes a severe neurodevelopmental disorder

*CACNB4* has previously been implicated in neurological disorders. To date, the nonsense variant p.(Arg482*) and the missense variant p.(Cys104Phe) in *CACNB4*, both in the heterozygous state, have been reported in patients with JME and episodic ataxia, respectively [15]. While the C-terminally truncated β$_4$ subunit altered Ca$_V$2.1 channel kinetics, the β$_4$-Cys104Phe mutant did not [15]. Genetic heterogeneity has been discussed to explain the absence of pathogenic *CACNB4* variants in large cohorts of patients with episodic ataxia [88]. However, the lack of replication studies raises some doubts on whether the identified variants in *CACNB4* have a

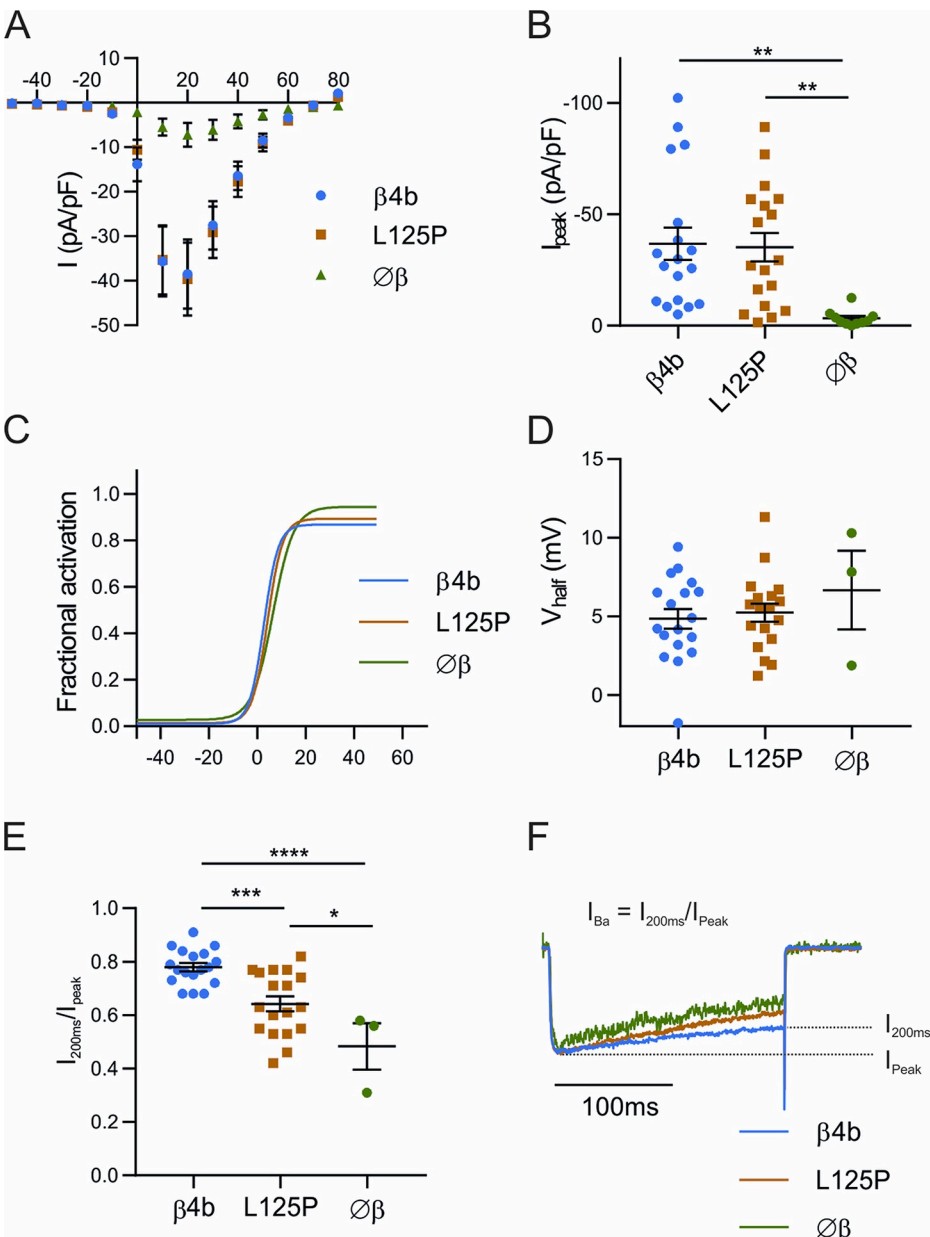

**Fig 7. Current properties of Ca$_V$2.1 calcium channels expressed in tsA201 cells with wild-type and the L125P mutant β$_{4b}$ subunit, plus α$_2$δ-1.** tsA201 cells were transfected with GFP-Ca$_V$2.1, α2δ-1, and β$_{4b}$-V5 (blue) or β$_{4b}$-L125P-V5 (orange) expressing plasmid or without a β subunit (green). Two to three days after transfection whole cell barium (15 mM) currents were recorded in response to 200ms test potentials increasing in 10 mV increments from -50 mV to +80 mV. The I/V curve and the scatter plot of Ipeak (A, B) show that co-expression of β$_{4b}$ as well as β$_{4b}$-L125P substantially increased current amplitudes. Fractional activation curves and scatter plot of the V1/2 (C, D) showed that the voltage sensitivity of the channel was not altered by co-expression of β$_{4b}$ or β$_{4b}$-L125P. The current inactivation estimated from the decline of the current at the end of the 200ms voltage step (E, F) showed that co-expression of β$_{4b}$ reduced current inactivation and that the mutant β$_{4b}$-L125P did so to a significantly lesser degree. I$_{peak}$: ANOVA F$_{(2, 43)}$ = 6.291; V$_{half}$: ANOVA F$_{(2, 36)}$ = 0.0591; Inactivation: ANOVA F$_{(2, 36)}$ = 15.56. Significance was calculated with Tukey post hoc test; * $P \leq 0.05$; ** $P \leq 0.01$; *** $P \leq 0.001$ **** $P \leq 0.0001$. Current properties were analysed on 18 cells for both β$_{4b}$ and β$_{4b}$-L125P. In the absence of a β subunit, the recorded currents were small (n = 10), restricting the analysis to only 3 cells. Results are expressed as Mean ± SEM.

major causal role in JME and/or episodic ataxia. The p.(Arg482*) variant is a rare nucleotide change and present in one individual of non-Finnish European descent in the gnomAD browser. The p.(Cys104Phe) variant has a minor allele frequency of 0.1% in the non-Finnish European population (131 in a total of 128,338 alleles in the gnomAD browser; no homozygotes). The o/e metrics in gnomAD indicate the *CACNB4* gene to be slightly intolerant to both non-synonymous (o/e: 0.55) and loss-of-function variants (o/e: 0.27). The recent finding of identical frequencies of ultra-rare variants in *CACNB4* in patients with a neurodevelopmental disorder with epilepsy compared with controls further suggests that heterozygous *CACNB4* variants are not associated with epilepsy [63]. In line with this, the parents of the two affected siblings we report here carry the p.(Leu126Pro) mutation in the heterozygous state and are healthy. Similarly, heterozygous *lethargic* mice do not show any abnormalities [89], and our co-expression experiment shows that the presence of mutant $\beta_{4b}$-L125P does not affect calcium channel association and nuclear targeting of wild-type $\beta_{4b}$ (S1 Fig). Taken together, combined evidence of exceptional high pathogenicity prediction scores for the p.Leu126Pro change, high conservation of leucine 126 within homologous protein domains, absence of the variant in population databases, and *CACNB4* as a gene slightly intolerant to missense variants suggests that the p.(Leu126Pro) mutation on both *CACNB4* alleles underlies the severe neurological phenotype in the two individuals reported here.

## Evidence for distinct pathomechanisms underlying the homozygous p. Leu126Pro mutation

**The p.Leu126Pro mutation impairs P/Q-type calcium channel functions.** The primary role of β subunits is that of an auxiliary subunit of voltage-gated calcium channels. The cytoplasmic β subunits bind to the pore-forming $Ca_V$ $\alpha_1$ subunits, promote their functional membrane expression, and modulate the channel gating properties [9]. Thus, the lack or a compromised function of a calcium channel β subunit might affect the function of neuronal calcium channels. As $\beta_4$ is prominently expressed throughout the brain, where it serves as the major β subunit partner of presynaptic P/Q-type calcium channels [5], defects in synaptic function and neuronal network activity would be expected as result of loss of $\beta_4$ function. Our results demonstrate that, as opposed to wildtype $\beta_{4b}$, stable association with $Ca_V1.2$ in the triads of dysgenic myotubes is abolished in the $\beta_{4b}$-L125P mutant (Fig 2), its targeting and incorporation into presynaptic boutons of cultured hippocampal neurons is abrogated (Fig 3), and it fails to stably complex with heterologously expressed $\alpha_1$ subunits (Fig 8). Together, these defects indicate that the L125P mutation abolishes or severely decreases the incorporation of $\beta_4$ into native calcium channel complexes, including the synaptic vesicle release machinery in CNS neurons. This finding is consistent with previous reports showing that the analogous amino acid substitution (L93P) in $\beta_{2a}$ disrupted the functionally important intra-molecular interaction between the SH3 and GK domains. When co-expressed with $Ca_V1.2$ in HEK cells mutated $\beta_{2a}$-L93P displayed a loss of functional interaction and modulation of the L-type calcium channel [31]; whereas when co-expressed with $Ca_V2.1$ in oocytes the $\beta_{2a}$-L93P mutant did not reduce P/Q-type currents but accelerated their inactivation at positive test potentials [32]. Similarly, we observed that the corresponding mutation in $\beta_{4b}$ had differential effects on membrane expression and modulation of its native channel partner $Ca_V2.1$. When co-expressed in tsA201 cells $\beta_{4b}$-L125P still augmented functional membrane expression of $Ca_V2.1$, but the effects of $\beta_{4b}$ on slowing inactivation were significantly blunted by the L125P mutation (Fig 7). Nevertheless, in the heterologous expression system the pore-forming $\alpha_1$ subunit did not efficiently co-precipitate with $\beta_{4b}$-L125P. Apparently, the L125P mutation weakens binding of $\beta_{4b}$ to the $\alpha_1$ subunit enough to abolish stable association and localization

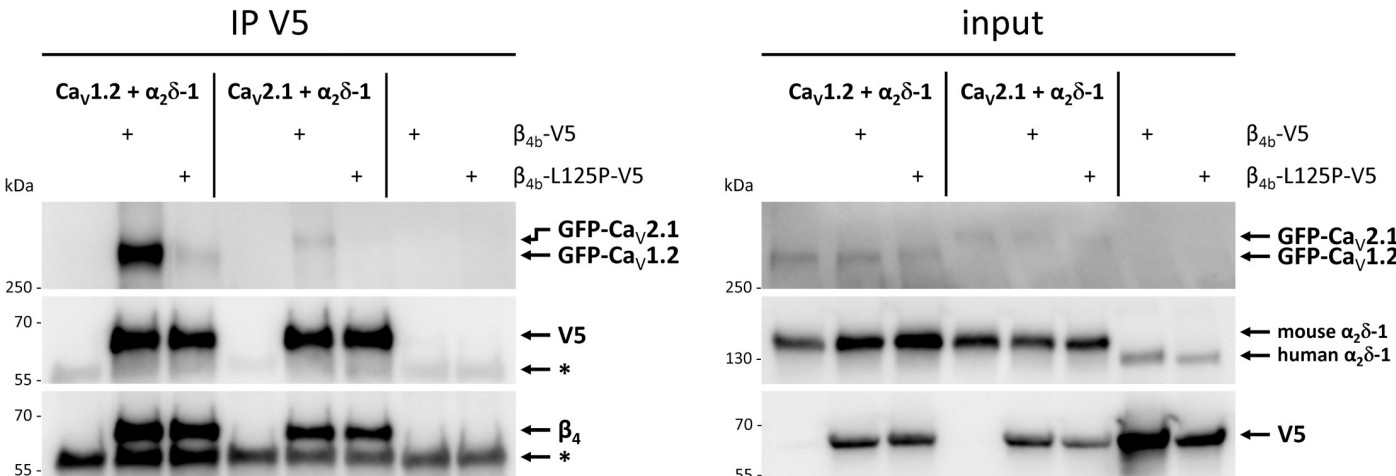

**Fig 8. The p.Leu125Pro mutation impairs complex formation of β$_{4b}$ with P/Q-type Ca$_V$2.1 and L-type Ca$_V$1.2 channels.** HEK293T cells were co-transfected with β$_{4b}$-V5 or β$_{4b}$-L125P-V5 expression construct, mouse α2δ-1 expression construct and GFP-Ca$_V$1.2 or GFP-Ca$_V$2.1 (α$_1$ subunits) expression construct as indicated. β$_{4b}$-V5 or β$_{4b}$-L125P-V5 was immunoprecipitated with V5-coupled Protein G dynabeads. As control, lysates from cells only transfected with β$_{4b}$-V5 or β$_{4b}$-L125P-V5 construct were incubated with an anti-normal mouse IgG antibody coupled to Dynabeads. Co-precipitated GFP-tagged α$_1$ subunit was detected by anti-GFP antibody. β$_{4b}$-V5 proteins were detected by anti-V5-HRP or anti-β$_4$ antibody. Ectopically expressed (mouse) and endogenous (human) α2δ-1 were detected by anti-α2δ-1 antibody (input, right panel). The band at ~55 kDa corresponds to the heavy chain of the IgG antibody and is marked by an asterisk in the immunoblots of the co-immunoprecipitation (IP V5; left panel). A representative blot of four (Ca$_V$1.2) and three (Ca$_V$2.1) independent experiments is shown.

in native channel complexes, while a high protein concentration upon overexpression in heterologous cells is still sufficient to support β$_{4b}$'s critical role in membrane trafficking [85–87].

Despite the severe consequences of the mutation on channel membrane trafficking and complex formation it is uncertain whether the p.Leu126Pro mutation leads to reduced P/Q-type calcium currents in patient cells and whether this is the main pathophysiology of the disease. The association of calcium channel β subunits with pore-forming α$_1$ subunits is highly promiscuous and neurons express multiple β isoforms [9, 27]. Therefore, it is possible that potential channel-dependent effects of the mutated β$_4$ subunits are compensated by the presence of other endogenously expressed β subunits.

Of note, the *lethargic* mouse model displays an autosomal recessive complex neurological disease that recapitulates many neurological anomalies present in the two affected individuals. Beside severe intellectual disability without speech, sister and brother developed epilepsy and had a movement disorder with athetoid and dystonic movements (Table 1). Similarly, *Cacnb4*-deficient mice develop ataxic gait with intermittent attacks of motor dysfunction resembling paroxysmal dyskinesia [89, 90], and in EEG recordings they display generalized cortical spike-wave discharges related to absence seizures [56, 91]. Adolescent mice experience a critical period with reduced body weight, renal cysts, and immunological anomalies that lead to increased mortality. Both of our patients have reduced weight, but they do not have immunological problems or renal anomalies. The most striking similarities were the cerebellar abnormalities in *lethargic* mice and the sib ship. While juvenile mutant mice did not show any changes in cerebellar morphology, the width of the cerebellar cortex was significantly reduced in adult *lethargic* mice [34]. In the affected female with the homozygous p.Leu126Pro mutation, brain imaging was normal at the age of 6 months, but showed cerebellar atrophy at the age of 8 years (S1 Text). In the boy, similar atrophies were already apparent at the age of 3 years and progressed over time (Fig 1). These data support the importance of β$_4$ for cerebellar development and maturation in mice and humans.

Finally, compound heterozygous mutations in the *CACNA1A* gene encoding the pore-forming $\alpha_{1A}$ subunit of the $Ca_V2.1$ voltage-gated calcium channel have been reported to cause epileptic encephalopathy with progressive cerebral atrophy, optic nerve atrophy, hypotonia, and severe developmental delay [92]. This phenotype shows significant overlap with that of the *lethargic* mice, mice with a selective deletion of P/Q-type channels in cerebellar Purkinje cells [22], as well as with that of the siblings reported here.

**The p.(Leu126Pro) mutation prevents activity-dependent nuclear targeting of $\beta_4$.**
Apart from the role as channel subunit, the $\beta_4$ splice variants $\beta_{4b}$ and $\beta_{4a}$ are involved in the activity-dependent regulation of gene expression [26, 28, 29]. Previously, the nuclear targeting defect of a $\beta_4$ mutant lacking 38 C-terminal amino acid residues ($\beta_{1-481}$) was attributed to cause juvenile myoclonic epilepsy [15, 29]. However, we observed normal nuclear targeting properties of the truncated $\beta_4(1–481)$ variant in three different cell systems [71]. In contrast, here we demonstrate that activity-dependent nuclear targeting of $\beta_{4b}$-L125P is abolished in cultured myotubes, cultured hippocampal neurons, and HEK293T cells (Figs 2, 4 and 6), consistent with previous data showing a cytoplasmic localization of $\beta_4$-L125P-EGFP [29]. Apparently, formation of a correct SH3-GK fold in $\beta_4$ is not only important for stable association of $\beta_4$ with the channel, but also for its import and retention in the nucleus. Because this nuclear targeting property is highly specific for particular $\beta_4$ splice variants [26], a deficiency in nuclear targeting properties or in the interaction with the transcriptional regulation machinery cannot be compensated by other $\beta$ isoforms.

**The p.Leu126Pro mutation abolishes complex formation between $\beta_4$ and TNIK.** While at this point the downstream signaling mechanisms and binding proteins affected by the loss of nuclear $\beta_{4b}$ targeting are elusive, our ongoing screen revealed a novel neuronal binding partner of $\beta_4$. Using co-immunoprecipitation, we demonstrate that wild-type $\beta_{4b}$ is in complex with TNIK, while the $\beta_{4b}$-L125P mutant could not be co-precipitated with TNIK (Fig 5B). TNIK is a neuronal serine-threonine kinase and scaffold protein that has been implicated in multiple neuronal processes including bidirectional glutamate receptor phosphorylation and the organization of nuclear complexes involved in the regulation of the neuronal transcription factor NeuroD1 [81]. As both $\beta_4$ subunit and TNIK are involved in Wnt/$\beta$-catenin signaling and transcriptional regulatory mechanisms [81, 83, 93], interaction of both proteins may regulate the Wnt pathway in neuronal cells and in neurogenesis in particular. In CNS synapses Tnik is concentrated in the postsynaptic density [81]. However, decreases in the frequency of miniature excitatory postsynaptic currents in *Tnik*[-/-] mice and Tnik's ability to regulate the number of synapses in *Caenorhabditis elegans* suggest a role of Tnik in presynaptic processes [81, 94]. Therefore, a possible functional link between TNIK and presynaptic calcium channels commonly regulating neurotransmitter release may exist. Future analysis will need to expose the function(s) of the TNIK-$\beta_{4b}$ protein complex in the neuronal system. In any case, the recent identification of biallelic loss-of-function variants in *TNIK* in individuals with intellectual disability [95] suggests a direct link between abrogated complex formation between TNIK and the $\beta_{4b}$-L125P mutant and impaired cognitive function in humans.

## Conclusions

Taken together, our study provides compelling evidence for the pathogenicity of the homozygous *CACNB4* missense mutation p.(Leu126Pro) and identifies three potential pathomechanisms which, separately or in combination, likely underlie the severe neurological disorder in the affected siblings. First, the p.Leu126Pro mutation impairs the formation of synaptic P/Q-type calcium channel complexes, second, it prevents activity-dependent nuclear targeting and

thus $\beta_4$-dependent nuclear functions, and third, it disturbs complex formation between $\beta_{4b}$ and the TRAF2 and NCK interacting kinase TNIK.

## Material and methods

### Study approval and ethics statement

All investigations were part of an ethically approved protocol (Ethics Committee of the Hamburg Medical Chamber; PV3802). Written informed consent was obtained for both affected siblings.

### Whole-exome sequencing and sequence data analysis

Genomic DNA was extracted from peripheral blood samples using standard procedures. We performed trio whole-exome sequencing (trio WES) with DNA samples of the male patient and both healthy parents as described before [36, 37]. Briefly, coding DNA fragments were enriched with a SureSelect Human All Exon 50Mb V5 Kit (Agilent), and captured libraries were then loaded and sequenced on a HiSeq2500 platform (Illumina). Reads were aligned to the human reference genome (UCSC GRCh37/hg19) using the Burrows-Wheeler Aligner (BWA, v.0.5.87.5), and detection of genetic variation was performed with SAMtools (v.0.1.18), PINDEL (v. 0.2.4t), and ExomeDepth (v.1.0.0). The functional impact of predicted amino acid substitutions was assessed by the pathogenicity prediction programs CADD (http://cadd.gs.washington.edu/score) [96], REVEL (https://sites.google.com/site/revelgenomics/downloads) [97], and M-CAP (http://bejerano.stanford.edu/MCAP/) [98].

### Variant validation

Sequence validation and segregation analysis of all candidate variants in the two affected siblings and their parents were performed by Sanger-sequencing. Primer pairs designed to amplify selected coding exons of the candidate genes (S1 Table) and exon-intron boundaries and PCR conditions are available on request. Amplicons were directly sequenced using the ABI BigDye Terminator Sequencing kit (Applied Biosystems) and an automated capillary sequencer (ABI 3500, Applied Biosystems). Sequence electropherograms were analyzed using the Sequence Pilot software (JSI Medical Systems).

### Plasmid information and cloning procedures

Cloning procedures for GFP-Ca$_V$1.2 (rabbit cDNA; X15593), GFP-Ca$_V$2.1 (rabbit cDNA; NM_001101693), p$\beta$A-$\beta_{1a}$-V5 (rabbit cDNA; M25514), p$\beta$A-$\alpha_2\delta$-1 (mouse cDNA; NM_009784) and p$\beta$A-$\beta_{4b}$-V5 (rat cDNA; L02315) were previously described [11, 27, 28, 99]. The L125P mutation was introduced by splicing by overlap extension (SOE) PCR. Briefly, nt 1–846 of $\beta_{4b}$ were PCR-amplified with overlapping primers introducing the point mutation T>C at nt position 374 in separate PCR reactions using p$\beta$A-$\beta_{4b}$-V5 as template. The two PCR products were then used as templates for a final PCR reaction with flanking primers to connect the nucleotide sequences. This fragment was then HindIII/EcoRV digested and cloned into the respective sites of p$\beta$A-$\beta_{4b}$-V5 yielding p$\beta$A-$\beta_{4b}$-L125P-V5. Sequence integrity of the newly generated construct was confirmed by sequencing (MWG Biotech).

### Cell culture and transfection

Skeletal myotubes of the homozygous dysgenic (mdg/mdg) cell line GLT [100] were cultured as previously described [100]. For immunofluorescence analysis the cells were plated on carbon/gelatin-coated coverslips in 35 mm dishes. At the onset of myoblast fusion, the cells were

transfected with 1 µg of plasmid DNA using FuGeneHD (Promega), according to the manufacturer´s instructions.

Low-density cultures of hippocampal neurons were obtained from 16.5–18 day old embryonic BALB/c mice of either sex as described previously [101–104]. Dissociated neurons were plated at a density of ~3500 cells/cm$^2$ on 18 mm glass coverslips (No 1.5; GML, Innsbruck, Austria) coated with poly-L-lysine (Sigma-Aldrich) in 60 mm culture dishes. After allowing the neurons to attach for 3–4 h, coverslips were transferred neuron-side down into a 60 mm culture dish containing a glial feeder layer. Maintenance of neurons and glia was done in serum-free neurobasal medium supplemented with Glutamax and B-27 (NBKO, all ingredients from Thermo Fisher Scientific). Plasmids were introduced into neurons at 6 DIV with Lipofectamine 2000-mediated transfection (Thermo Fisher Scientific) as described previously [103]. For co-transfection of pβA-eGFP plus pβA-β$_{4b}$-V5 or pβA-β$_{4b}$-L125P-V5 1 µg total DNA was used at a molar ratio of 1:1. Cells were processed for immunostaining experiments between 27–35 DIV. For analyzing the activity dependent nuclear targeting of β$_{4b}$ TTX (1 µM) was added to the culture medium and neurons were incubated overnight (12 h).

HEK293T and tsA201 cells were cultured in Dulbecco´s modified Eagle medium (DMEM, Thermo Fisher Scientific) supplemented with 10% (v/v) fetal bovine serum (FBS; Merck) and penicillin-streptomycin (100 U/ml and 100 µg/ml, respectively; Thermo Fisher Scientific) and incubated at 37˚C in a humidified atmosphere with 5% CO$_2$. HEK293T cells were transiently transfected with 1 µg (immunocytochemistry) or 5 µg of plasmid DNA (co-immunoprecipitation) using TurboFect (Thermo Fisher Scientific) for 6 h according to the protocol provided and cultured in DMEM overnight.

## Immunocytochemistry

**Dysgenic myotubes**.   Cells were immunostained at day 9–10, as described in [26, 77].

**Cultured hippocampal neurons**.   Immunolabeling was performed as described previously [71, 103, 104]. Höchst 33342 dye (~5 µg/ml) was applied to the immunostained neurons for 30 sec in PBS/BSA/Triton to label the nuclei.

**HEK293T cells**.   Coverslips were coated with 10 µg/ml collagen type I (Merck) in PBS for 1 hour at room temperature. Excess of collagen was removed, HEK293T cells were seeded on coverslips and transfected with expression constructs. Subsequently, cells were rinsed with PBS, fixed with 4% paraformaldehyde (Merck) in PBS and washed three times with PBS. After treatment with permeabilization/blocking solution (2% BSA, 3% goat serum, 0.5% Nonidet P40 in PBS), cells were incubated in antibody solution (3% goat serum and 0.1% Nonidet P40 in PBS) containing appropriate primary antibodies. Cells were washed with PBS and incubated with Fluorophore-conjugated secondary antibodies in antibody solution. After extensive washing with PBS cells were embedded in ProLong Diamond Antifade Mountant with DAPI (Thermo Fisher Scientific) on microscopic slides.

## Antibodies

The following primary antibodies (Thermo Fisher Scientific) were used for immunocytochemistry: mouse anti-V5 (1:400 –Figs 2–4; 1:300 –Fig 6; R960-25, Thermo Fisher Scientific), rabbit anti-PPP2R5D (1:500; A301-098A, Bethyl Laboratories Inc.) and rabbit anti-GFP (1:10,000; A6455, Thermo Fisher Scientific). Secondary antibodies (Thermo Fisher Scientific) were used at 1:4,000 (Figs 2–4) or 1:1,000 (Fig 6): Alexa Fluor 488 goat anti-mouse (A11001), Alexa Fluor 488 goat anti-rabbit (A11008), Alexa Fluor 594 goat anti-mouse (A11032), Alexa Fluor 546 goat anti-rabbit (A11010) and Alexa Fluor 647 goat anti-rabbit antibodies (A32733).

For immunoblotting and immunoprecipitation: mouse anti-CACNA2D1 ($\alpha_2\delta$-1) (20A) antibody (1:500, MA3-921, Thermo Fisher Scientific), mouse anti-Ca$_V$β$_4$ calcium channel antibody (1:500; 75–054, NeuroMab), mouse anti-GFP antibody (1:5000; 902601, BioLegend), anti-normal mouse IgG (1:100; 12–371, Merck Millipore) and anti-normal rabbit IgG antibody (1:100; 12–370, Merck Millipore), rabbit anti-PPP2R5D antibody (IP 1:25; WB 1:2,500; A301-098A, Bethyl Laboratories Inc.), rabbit anti-TNIK antibody (IP 1:50; WB 1:1,000; 32712, Cell Signaling Technologies), mouse anti-V5 (1:125; R960-25, Thermo Fisher Scientific), mouse anti-V5 tag horseradish peroxidase (HRP)-coupled antibody (1:5,000; R961-25, Thermo Fisher Scientific). Secondary HRP-coupled anti-rabbit (1:5,000; NA934V) and anti-mouse (1:5,000; NA931V) antibodies were from GE Healthcare.

## Microscopy

Preparations of dysgenic myotubes were analyzed on an Axioimager microscope (Carl Zeiss) using a 63x 1.4 NA objective. 14-bit images were acquired using Metamorph software (Universal Imaging) connected to a cooled CCD camera (SPOT, Diagnostic Instruments). Figures were arranged in Adobe Photoshop and, where necessary, linear adjustments were performed to correct black level and contrast. Immunostained hippocampal neurons were observed with a BX53 microscope (Olympus) using a 60× 1.42 NA oil-immersion objective lens and fourteen-bit gray-scale images were recorded with a cooled CCD camera (XM10, Olympus) using cellSens Dimension software (Olympus). Images were analyzed with MetaMorph software (Molecular Devices) or ImageJ/Fiji [105] as described below. Figures were assembled in Adobe Photoshop CS6 and linear adjustments were done to correct black level and contrast. HEK293T cells were examined in epifluorescence mode of an Olympus cell tool TIRFM system (Olympus) equipped with a 60x oil immersion objective lens, and pictures were taken of representative cells to visualize subcellular localization of endogenous PPP2R5D and V5-tagged proteins.

## Quantification of neuronal β$_{4b}$ expression

Analysis of β$_{4b}$ wildtype and mutant expression was performed with ImageJ/Fiji [105] as follows. The axon initial segment was identified based on morphological criteria in the eGFP image, and a ~30 μm long line was traced along the axon hillock and a background region was selected near the axon hillock. Similarly, a ~30 μm long line was traced along the distal axon (> 250 μm from the cell soma) in the eGFP image, and a background region was selected accordingly. ROIs were transferred from the eGFP images to the corresponding anti-V5 images, and average eGFP and V5 intensities were automatically recorded. Average axon hillock and distal axon labeling intensity of each cell was divided by the corresponding average background intensity, and hence labeling intensities are expressed as fold-expression above background. Statistical analysis was performed using MS Excel.

## Co-clustering and nuclear targeting analysis

Cultures labeled with anti-GFP (GFP-Ca$_V$1.2) and anti-V5 (β-V5) were systematically screened for transfected, well differentiated myotubes based on the clustered GFP staining of the calcium channel. After switching to the red channel, the co-clustering and the nuclear staining of the β subunits were analyzed. Nuclear targeting of the β subunit rated positive when the fluorescence intensity of any nuclei in the myotube was above that of the cytoplasm. The degree of nuclear targeting was determined by calculating the nucleus/cytoplasm ratio of the background substracted anti-V5 fluorescence intensity using Metamorph. The degree of nuclear targeting in cultured hippocampal neurons was analyzed employing a custom

programmed Metamorph Journal as previously described [71]. Results are expressed as mean ± SEM unless otherwise indicated. All data were organized in Microsoft Excel and analyzed using ANOVA or 2-way ANOVA in GraphPad or Sigmaplot. "N" refers to the number of independent experiments and "n" to the number of individual cells analyzed.

## Co-immunoprecipitation and immunoblotting

Co-immunoprecipitations were performed with magnetic Dynabeads Protein G (Thermo Fisher Scientific). Therefore, 2.5 µg of anti-PPP2R5D antibody (Fig 5A), anti-TNIK antibody (1:50 dilution; Fig 5B), or anti-V5 antibody (1:125 dilution; Fig 8) was bound to Dynabeads on a rotator for 10 min at room temperature followed by a washing step with co-immunoprecipitation buffer [50 mM Tris-HCl pH 8, 120 mM NaCl, 1 mM EDTA, 0.5% Nonidet P40; supplemented with complete Mini Protease Inhibitors (Roche) and PhosphoStop (Roche)]. Cells were lysed in 500 µl co-immunoprecipitation buffer for 10 min at 4˚C, and cell debris was cleared by centrifugation for 10 min. After removing an aliquot (total cell lysate), 250 µl of the remaining supernatant was incubated with the antibody-bound Dynabeads for 2 h at 4˚C on a rotator. Subsequently, the Dynabeads were pelleted and washed four times with co-immunoprecipitation buffer. The bound target proteins were eluted by resuspending the beads in 50 µl 1x sample buffer, separated on SDS-PAGE under denaturing conditions, and transferred to PVDF (polyvinylidene fluoride) membranes (Bio-Rad). Membranes were blocked followed by incubation with the indicated primary antibody overnight at 4˚C and by HRP (horseradish peroxidase)-linked secondary antibodies at room temperature for 1 h. Chemiluminescent western blots were digitally imaged using a ChemiDoc MP (Bio-Rad).

## Electrophysiology and data analysis

tsA201 cells were transfected using the calcium phosphate transfection method as previously described [106] with GFP-$Ca_V$2.1, $\alpha_2\delta$-1 and either $\beta_{4b}$-V5, $\beta_{4b}$-L125P-V5 or no $\beta$ subunit. 24h after the transfection, cells were replated on 35mm culture dishes coated with poly-L-lysine and kept in 5% $CO_2$ at 30˚C; electrophysiological recordings were performed during the following two days. Currents were recorded using the whole-cell patch clamp technique in voltage-clamp mode using an Axopatch 200B amplifier (Axon Instruments). Patch pipettes (borosilicate glass; Sutter Instrument) had resistances between 1.8 and 3.5 MΩ and were filled with 144.5 mM Cs-Cl, 1 mM MgCl$_2$, 10 mM HEPES, 10 mM Cs-EGTA, and 4 mM Na2-ATP (pH 7.4 with Cs-OH). Bath solution contained 15 mM BaCl$_2$, 150 mM choline chloride, 1mM MgCl$_2$, and 10 mM HEPES (pH 7.4 with tetraethylammonium hydroxide). Data acquisition and command potentials were controlled by Clampex software (v10.6; Axon Instruments); analysis was performed using Clampfit 10.5 (Axon Instruments) and Sigma-Plot 8.0 (SPSS Science) software. The current-voltage relationships were obtained by applying a 200 ms-long square pulse from -50mV to +80mV in 10 mV steps, starting from a holding potential of -80 mV. The I/V curves were fitted according to I = $G_{max}$ ·(V−$V_{rev}$) / 1(1+exp (− (V−$V_{0.5}$) / $k_a$)), where $G_{max}$ is the maximum conductance of the slope conductance, $V_{rev}$ is the extrapolated reversal potential of the calcium current, $V_{0.5}$ is the potential for half maximal conductance, and $k_a$ is the slope factor. The conductance was calculated using G = (−I$^*$1000) / (V−$V_{rev}$), and its voltage dependence was fitted according to a Boltzmann distribution: G = $G_{max}$ / (1+exp (− (V−$V_{0.5}$) / $k_a$)). Channel inactivation was quantified by calculating the ratio between residual current at the end of the 200ms sweep and at the maximum ($I_{res200}$). All quantitative data are expressed as mean ± SEM. Statistical significance was determined by one-way ANOVA followed by Tukey post-hoc analysis, as indicated using GraphPad Prism. Significance was set to $p < 0.05$.

## Supporting information

**S1 Text. Clinical report of the patients.**
(DOCX)

**S1 Fig. Differential channel association and nuclear targeting of wildtype and L125P mutant β₄ᵦ co-expressed in dysgenic myotubes.** Dysgenic (Ca$_V$1.1-null) muscle cells were transfected with Ca$_V$1.2 together with both β$_{4b}$(wt)-GFP plus β$_{4b}$-L125P-V5 to mimic the situation in heterozygous carriers of the *CACNB4* variant, and immunolabeled with anti-GFP (green) and anti-V5 (red). (A) The β$_{4b}$(wt)-GFP displayed the typical clustered distribution, resembling the β$_{4b}$ subunit incorporated into calcium channel complexes (examples indicated by arrow heads). In contrast, the co-expressed mutant β$_{4b}$-L125P-V5 was evenly dispersed throughout the cytoplasm, indicating its failure to associate with the pore-forming Ca$_V$1.2 subunit. (B and C) In quiescent cells (immature myotubes (B) or myoblasts (C)) that displayed nuclear targeting of the wildtype β$_{4b}$(wt)-GFP subunit, the mutant β$_{4b}$-L125P-V5 failed to accumulate in the nuclei (examples indicated by arrows). N = 5. Scale bars, 10 μm.
(TIF)

**S1 Table. *In silico* pathogenicity prediction, minor allele frequency, and associated OMIM phenotypes of shared biallelic variants in patients 1 and 2.** Trio-exome data were filtered for potentially pathogenic *de novo* variants absent in the general population (dbSNP138, 100 Genomes Project, Exome Variant Server, ExAC Browser, and gnomAD Browser) and rare biallelic variants with minor allele frequency (MAF) <0.1% and no homozygous carriers in the aforementioned databases. MetaDome web server (https://stuart.radboudumc.nl/metadome) combines resources and information from genomics and proteomics to improve variant interpretation by transposing this variation to homologous protein domains. It visualizes meta-domain information and gene-wide profiles of genetic tolerance [70]. The constraint score shown in gnomAD is the ratio of the observed/expected (o/e) number of missense variants in that gene. The functional impact of the identified variants was predicted by the Combined Annotation Dependent Depletion (CADD) tool, the Rare Exome Variant Ensemble Learner (REVEL) scoring system, and the Mendelian Clinically Applicable Pathogenicity (M-CAP) Score. CADD is a framework that integrates multiple annotations in one metric by contrasting variants that survived natural selection with simulated mutations. Reported CADD scores are phred-like rank scores based on the rank of that variant's score among all possible single nucleotide variants of hg19, with 10 corresponding to the top 10%, 20 at the top 1%, and 30 at the top 0.1%. The larger the score the more likely the variant has deleterious effects; the score range observed here is strongly supportive of pathogenicity, with all observed variants ranking above ~99% of all variants in a typical genome and scoring similarly to variants reported in ClinVar as pathogenic (~85% of which score >15) [96]. REVEL is an ensemble method predicting the pathogenicity of missense variants with a strength for distinguishing pathogenic from rare neutral variants with a score ranging from 0–1. The higher the score the more likely the variant is pathogenic [97]. M-CAP is a classifier for rare missense variants in the human genome, which combines previous pathogenicity scores (including SIFT, Polyphen-2, and CADD), amino acid conservation features and computed scores trained on mutations linked to Mendelian diseases. The recommended pathogenicity threshold is >0.025 [98]. Chr., chromosome; DFNB3: Deafness, autosomal recessive 3; EA5: Episodic ataxia, type 5; EIG9: Epilepsy, idiopathic generalized, susceptibility to, 9; EJM6: Epilepsy, juvenile myoclonic, susceptibility to, 6; MAF, minor allele frequency; RP84: Retinitis pigmentosa 84; –, not available.
(PDF)

**S2 Table. Current parameters.** Data are expressed as mean value ± SEM.
(PDF)

## Acknowledgments

We thank the family for the participation in this study. We further thank R. Stanika, K. Heinz and M. Heitz at the Division of Physiology, Medical University Innsbruck, D. Zorndt at the Institute of Human Genetics, University Medical Center Hamburg-Eppendorf, and the staff of the Microscopy Imaging Facility at the University Medical Center Hamburg-Eppendorf (UMIF) for technical support.

## Author Contributions

**Conceptualization:** Pierre Coste de Bagneaux, Leonie von Elsner, Marta Campiglio, Gerald J. Obermair, Bernhard E. Flucher, Kerstin Kutsche.

**Data curation:** Pierre Coste de Bagneaux, Leonie von Elsner, Gerald J. Obermair.

**Funding acquisition:** Gerald J. Obermair, Bernhard E. Flucher, Kerstin Kutsche.

**Investigation:** Pierre Coste de Bagneaux, Leonie von Elsner, Tatjana Bierhals, Marta Campiglio, Jessika Johannsen, Maja Hempel.

**Supervision:** Gerald J. Obermair, Bernhard E. Flucher, Kerstin Kutsche.

**Validation:** Leonie von Elsner, Kerstin Kutsche.

**Writing – original draft:** Pierre Coste de Bagneaux, Leonie von Elsner, Marta Campiglio, Jessika Johannsen, Gerald J. Obermair, Maja Hempel, Bernhard E. Flucher, Kerstin Kutsche.

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
