## [Decision Letter · Decision Letter 0]

11 Oct 2019

Dear Dr Kutsche,

Thank you very much for submitting your Research Article entitled 'A homozygous missense variant in CACNB4 encoding the auxiliary calcium channel beta4 subunit causes a severe neurodevelopmental disorder and impairs channel and non-channel functions' to PLOS Genetics. Your manuscript was fully evaluated at the editorial level and by independent peer reviewers. The reviewers appreciated the attention to an important problem, but raised some substantial concerns about the current manuscript. Based on the reviews, we will not be able to accept this version of the manuscript, but we would be willing to review again a much-revised version. We cannot, of course, promise publication at that time

We have received reviews on your submissions “A homozygous missense variant in CACNB4 encoding the auxiliary calcium channel beta4 subunit causes a severe neurodevelopmental disorder and impairs channel and non-channel functions” that ranged from enthusiastic to skeptical and that are listed below.

While the functional studies are thorough and well executed, a general question raised by the reviewers was whether the homozygous variants in CACNB4 was in fact causative in the family reported in your study, given that it represents a missense variant in a consanguineous population that is not well represented in population databases.

While it is often impossible to provide independent genetic evidence for ultra-rare genetic diseases as identifying further individuals with bi-allelic changes in this genes may be impossible, this uncertainty may be addressed in the revised version and you and your team may consider querying available genomic resources (DDD, epilepsy/autism datasets in dbgap) whether additional individuals with bi-allelic CACNB4 variants have been identified.

Given the comments of the reviewers, we feel that your manuscript may require significant changes prior to acceptance and would ask you to submit a major revision.

If you decide to revise the manuscript for further consideration at PLOS Genetics, please aim to resubmit within the next 60 days, unless it will take extra time to address the concerns of the reviewers, in which case we would appreciate an expected resubmission date by email to plosgenetics@plos.org.

[LINK]

We are sorry that we cannot be more positive about your manuscript at this stage. Please do not hesitate to contact us if you have any concerns or questions.

Yours sincerely,

Ingo Helbig, MD

Guest Editor

PLOS Genetics

Gregory Barsh

Editor-in-Chief

PLOS Genetics

Reviewer's Responses to Questions

**Comments to the Authors:**

Reviewer #1: review is uploaded as an attachment (5295 characters).

Reviewer #2: The authors provide a very nice article which solves a long debate about the role of variants in CACNB4 to cause disorders of the nervous system in humans. Although they only report about one family with two recessively affected sibs, their functional work is highly comprehensive and demonstrates the pathological role of the homozygous variant undoutedly. Interestingly, a simple expression in HEK or tsA201 cells of the mutant vs. WT subunit together with alpha1 and alpha2delta subunits would not have revealed this pathological role. But the extensive studies in neurons, myotubes and other cells including biochemistry and immunohistochemistry reveal the defective interaction of the mutant CACNB4 subunit with various other proteins disrupting both the electrophysiological function of the P/Q- or L-type Ca channel complex and the interaction with nuclear targeting and signaliing. These reeults are highly convincing and well performed. I have no critique or suggestions for changes.

The study is highly important, since it terminates a long debate about eventually disease-causing heterozygous variants which becomes very unlikely due to their study (although a smaller contribution of disease-modifying effects of het variants can still not - may be never - be excluded).

Reviewer #3: In this manuscript the authors describe a rare homozygyous variant in CACNB4 that is found in 2 siblings with a severe neurodevelopmental disorder. The clinical, genetic and functional data are of a high standard. The manuscript is generally well written although it could be tightened up a little—especially the discussion but also bits of the results. The authors provide strong evidence that the CACNB4 disrupts some physiological processes. However, I am not completely convinced that the leu126Pro is the cause of the severe neurodevelopmental phenotype seen in the siblings.

First, there are 6 candidate variants that could be responsible for disease. The authors systematically rule out those other than CACNB4 based on a combination of in silico, animal model and clinical data. The case for exclusion is not that strong in my opinion with each of the ‘exclusion methods’ frequently shown to be wanting. For example, in silico analysis is particularly fraught. While I accept that completing functional analysis on all is not practical it remains the best way of ruling out these variants (although even this approach has its issues).

The functional impact of the leu126Pro variant goes someway in arguing that it is potentially pathogenic. However, all the assays that showed differences relate to binding and/or trafficking. The biological consequences of these is not clear to me. It would be good to see a tractable output that had a biological relevance and point to a pathomechanism. The only assay the specifically looks at a functional output that is tractable is the electrophysiology in which the variant has no impact.

Perhaps the most compelling way to confirm pathogenicity is the engineering of a rodent model. With CRISPR technology this is no longer prohibitive.

In summary, although the manuscript contains very solid data it lacks compelling evidence that the studied CACNB4 leu126Pro variant is causing disease.

Reviewer #4: In this study, de Bagneaux and colleagues report a homozygous mutation in CACNB4 identified in two siblings with severe intellectual disability, blindness, epilepsy, movement disorder and cerebellar atrophy. They report functional analysis of the identified variant using the expression in cultured hippocampal neurons, tSA cells and cultured myotubes and suggest that the mutant beta 4 subunit shows reduced incorporation in the presynaptic calcium complexes, absence of nuclear targeting and abolished interaction with a neuronal kinase. However, coexpression of the mutant with the Cav2.1 in tsA201 cells affected the calcium currents in a similar way as the beta 4 wild type. The authors suggest that their data provide evidence for the pathogenicity of the detected variant and corroborate the role of CACNB4 in human disease.

This is a well written and structured manuscript providing elaborate analysis and some interesting insights into the function of the beta 4 subunit and its potential role in the severe neurodevelopmental disorders. The major concern in my view is that while L125P variant seems to affect localization and interactions of the beta 4 subunit with other proteins, there is no clear indication in this manuscript how this may lead to the severe disease seen in the two patients. This question can be addressed in several ways that could corroborate the conclusions of the study. Firstly, since the variant is present in unaffected parents, a comparison to the heterozygous expression would be a very useful control. Furthermore, given the discrepancy between expression in heterologous systems and neurons, a functional assay pointing at the altered neuronal/synaptic activity would be indicative of the underlying disease mechanism. Lastly, any insights into the changes of the gene expression resulting from the altered nuclear translocation or interaction with TNIK would be useful to understand the proposed effect of the variant.

The paper would also be improved by providing more consistent quantification of morphological and biochemical data. For instance, in Fig 3, have the authors considered neurite tracing and quantification? Also, axon hillock expression (Fig 4) should be quantified.

Fig 5 shows that the input lysates in both experiments had less mutant protein to start with. Is the reduced expression of the L126P part of the mechanism?

**Have all data underlying the figures and results presented in the manuscript been provided?**

Reviewer #1: Yes

Reviewer #2: Yes

Reviewer #3: Yes

Reviewer #4: None

PLOS authors have the option to publish the peer review history of their article (what does this mean?). If published, this will include your full peer review and any attached files.

Reviewer #1: Yes: Géza Berecki, PhD

Senior Research Fellow

Ion Channels and Human Diseases Laboratory

The Florey Institute of Neuroscience and Mental Health

The University of Melbourne VIC 3010

geza.berecki@florey.edu.au

geza.berecki@unimelb.edu.au

Reviewer #2: Yes: Holger Lerche

Reviewer #3: No

Reviewer #4: No

---

## [Decision Letter · Decision Letter 1]

23 Jan 2020

Dear Dr Kutsche,

We are pleased to inform you that your manuscript entitled "A homozygous missense variant in CACNB4 encoding the auxiliary calcium channel beta4 subunit causes a severe neurodevelopmental disorder and impairs channel and non-channel functions" has been editorially accepted for publication in PLOS Genetics. Congratulations!

Yours sincerely,

Ingo Helbig, MD

Guest Editor

PLOS Genetics

Gregory Barsh

Editor-in-Chief

PLOS Genetics

Comments from the reviewers (if applicable):

Reviewer's Responses to Questions

**Comments to the Authors:**

Reviewer #1: The authors have addressed all my comments and suggestions. The manuscript has been significantly improved by the revision.

Reviewer #3: The authors should be commended on the lengths they have gone to in order to find a plausible mechanism. Softening the conclusion is important in light of the lack of mechanism for excitability, which the authors have done.

Reviewer #4: This was a solid and thorough study to begin with, and the authors have now performed additional experiments and addressed the concerns raised by reviewers. While some critical questions remain to be answered in future, I do not have further concerns regarding this manuscript.

**Have all data underlying the figures and results presented in the manuscript been provided?**

Reviewer #1: Yes

Reviewer #3: Yes

Reviewer #4: No: The values behind the means, standard deviations and other measures reported should be provided as part of 'minimal data set'.

PLOS authors have the option to publish the peer review history of their article (what does this mean?). If published, this will include your full peer review and any attached files.

Reviewer #1: No

Reviewer #3: No

Reviewer #4: No

**Data Deposition**

http://datadryad.org/submit?journalID=pgenetics&manu=PGENETICS-D-19-01072R1

**Press Queries**

---

## [Editor Report · Acceptance letter]

4 Mar 2020

PGENETICS-D-19-01072R1 

A homozygous missense variant in CACNB4 encoding the auxiliary calcium channel beta4 subunit causes a severe neurodevelopmental disorder and impairs channel and non-channel functions 

Dear Dr Kutsche, 

We are pleased to inform you that your manuscript entitled "A homozygous missense variant in CACNB4 encoding the auxiliary calcium channel beta4 subunit causes a severe neurodevelopmental disorder and impairs channel and non-channel functions" has been formally accepted for publication in PLOS Genetics! Your manuscript is now with our production department and you will be notified of the publication date in due course.

With kind regards,

Kaitlin Butler

PLOS Genetics

On behalf of:
